# Invasive *Salmonella* exploits divergent immune evasion strategies in infected and bystander dendritic cell subsets

Anna Aulicino[1,2], Kevin C. Rue-Albrecht [3], Lorena Preciado-Llanes[1,2], Giorgio Napolitani[1], Neil Ashley[4], Adam Cribbs[5], Jana Koth[6], B. Christoffer Lagerholm[6], Tim Ambrose[1,2], Melita A. Gordon[7,8], David Sims[5] & Alison Simmons [1,2]

Non-typhoidal *Salmonella* (NTS) are highly prevalent food-borne pathogens. Recently, a highly invasive, multi-drug resistant *S.* Typhimurium, ST313, emerged as a major cause of bacteraemia in children and immunosuppressed adults, however the pathogenic mechanisms remain unclear. Here, we utilize invasive and non-invasive *Salmonella* strains combined with single-cell RNA-sequencing to study the transcriptome of individual infected and bystander monocyte-derived dendritic cells (MoDCs) implicated in disseminating invasive ST313. Compared with non-invasive *Salmonella*, ST313 directs a highly heterogeneous innate immune response. Bystander MoDCs exhibit a hyper-activated profile potentially diverting adaptive immunity away from infected cells. MoDCs harbouring invasive *Salmonella* display higher expression of *IL10* and *MARCH1* concomitant with lower expression of *CD83* to evade adaptive immune detection. Finally, we demonstrate how these mechanisms conjointly restrain MoDC-mediated activation of *Salmonella*-specific CD4+ T cell clones. Here, we show how invasive ST313 exploits discrete evasion strategies within infected and bystander MoDCs to mediate its dissemination in vivo.

[1] MRC Human Immunology Unit, MRC Weatherall Institute of Molecular Medicine, University of Oxford, Oxford OX3 9DS, UK. [2] Translational Gastroenterology Unit, John Radcliffe Hospital, Headington, Oxford OX3 9DU, UK. [3] Kennedy Institute of Rheumatology, Nuffield Department of Orthopaedics, Rheumatology and Musculoskeletal Sciences, University of Oxford, Headington, Oxford OX3 7FY, UK. [4] MRC Molecular Haematology Unit, MRC Weatherall Institute of Molecular Medicine, University of Oxford and BRC Blood Theme, NIHR Oxford Biomedical Centre, Oxford OX3 9DS, UK. [5] MRC WIMM Centre for Computational Biology, MRC Weatherall Institute of Molecular medicine, University of Oxford, Oxford OX3 9DS, UK. [6] MRC Human Immunology Unit and Wolfson Imaging Centre, MRC Weatherall Institute of Molecular Medicine, University of Oxford, Oxford OX3 9DS, UK. [7] Institute of Infection and Global Health, University of Liverpool, 8 W Derby St, Liverpool L7 3EA, UK. [8] Malawi-Liverpool Wellcome Trust Clinical Research Programme, Blantyre, Malawi. These authors contributed equally: Anna Aulicino, Kevin C. Rue-Albrecht  Correspondence and requests for materials should be addressed to A.S. (email: alison.simmons@ndm.ox.ac.uk)

Non-typhoidal *Salmonella* (NTS) are among the most common food-borne pathogens, which cause 93 million cases of gastroenteritis each year, including 155,000 deaths[1,2]. The most frequently reported, *Salmonella enterica* serovar Typhimurium (*S.* Typhimurium), causes localized, self-limiting gastroenteritis and infects both humans and animals[3]. Recently, a highly invasive multi-drug resistant *S.* Typhimurium of a distinct multi-locus sequence type (MLST), ST313, has emerged. Its prevalence across sub-Saharan Africa became a major cause of lethal bacteraemia in young malnourished anaemic children and individuals affected by concomitant diseases such as malaria or HIV[4,5]. *S.* Typhimurium ST313 pathovar D23580 (STM-D23580) is a representative bloodstream clinical isolate, which exhibits genomic degradation resembling the human-restricted pathogen *S.* Typhi[4]. How ST313 usurps mucosal immune defences to achieve an invasive phenotype in susceptible individuals is unclear. In particular, whether it usurps dendritic cell (DC) functions to avoid adaptive immunity has not been elucidated.

DCs are key professional antigen-presenting cells (APCs) of the immune system. Activating pattern recognition receptors (PRRs) expressed in DCs stimulates their maturation and induces antigen-specific immunity[6,7]. The primed immune response critically depends on the specific signalling context triggered by invading bacteria. Thus, the ability of pathogenic bacteria to subvert DC functions and prevent T cell recognition contributes to their survival and dissemination within susceptible hosts and cell subsets. *Salmonella* can modulate DC functions[8–10]. However, it remains unclear whether individual DCs differentially recognize genetically similar *S.* Typhimurium strains and how differential innate immune sensing produces distinct functional outcomes. Furthermore, the mechanisms of action that facilitate ST313 evasion strategies in specific DCs in human disease remain unclear.

Transcriptomics approaches are utilized to survey and identify host cellular signalling pathways in response to intracellular bacteria[11]. Earlier analyses of bulk samples amalgamated the gene expression profiles of millions of cells and did not account for the phenotypic diversity between individual cells. In contrast, single-cell RNA-sequencing (scRNA-seq) has emerged as a powerful tool to profile previously inaccessible cell-to-cell transcriptional variability on a genome-wide scale[12–14]. Pioneering scRNA-seq analyses of lipopolysaccharide (LPS)-stimulated murine bone-marrow-derived DCs revealed an unexpected degree of bimodal expression among immune genes exposed to uniform pathogen-associated molecular patterns (PAMPs), otherwise detected at seemingly consistent high levels as average population measurements[15]. More recently, scRNA-seq revealed considerable heterogeneity in responses of individual murine macrophages to variations in intracellular *Salmonella* growth[16].

Here, we combine fluorescent-activated cell sorting (FACS) and scRNA-seq to survey the transcriptome of individual human MoDCs challenged with invasive or non-invasive *Salmonella*. Using this system, we measure the transcriptional profile with high sensitivity and full-length gene coverage. We distinguish infected MoDCs, in which *Salmonella* persists and adapts to the host, from neighbouring cells, either stimulated by bacterial PAMPs or that have engulfed and processed bacterial moieties. We elucidate the mechanisms of action that ST313 utilizes to disseminate in specific MoDC subsets. Together, our scRNA-seq results reveal the mechanisms of cell-intrinsic host adaption exploited by *Salmonella* ST313. These mechanisms, in conjunction with bystander hyper-activation, provide insight for its invasive behaviour in immunocompromised hosts.

## Results

**Single-cell RNA-sequencing of challenged human MoDCs.** To profile the transcriptional response of individual human MoDCs infected with bacteria and compare it with that of bystander cells, we labelled STM-LT2 and STM-D23580 with CellTrace[TM] Violet Cell Proliferation dye prior to infection (Fig. 1a and Supplementary Figure 1). MoDCs that engulfed *Salmonella* could be identified by their emitted Violet fluorescence, while bystander MoDCs exhibited no Violet signal (Supplementary Figure 2). Internalization of both bacterial strains was also confirmed by confocal microscopy using a specific anti-*Salmonella* antibody (Supplementary Figure 3).

We confirmed the presence of live *Salmonella* within infected cells by sorting MoDCs by their fluorescence phenotype and enumerating intracellular bacteria after cell lysis. Infected cells showed constant numbers of intracellular bacteria over time, while no or very few viable bacteria were recovered from bystander MoDCs (Supplementary Figure 4). STM-LT2 and STM-D23580 demonstrated equivalent abilities to survive and multiply within MoDCs, and no significant differences were observed in the number of CFU between bacterial strains at each time point (Supplementary Figure 5A). The percentage of uptake and survival was also comparable for both strains (Supplementary Figure 5B and 5C). Moreover, no significant differences were observed in the viability of MoDCs infected with the two bacterial strains during the course of the infection (Supplementary Figure 5D).

Individual infected or bystander MoDCs and uninfected MoDCs from mock-treated cultures were isolated by FACS sorting at 2, 4 and 6 h after infection. We then performed scRNA-seq on single sorted MoDCs according to the Smart-seq2 protocol[17] (Fig. 1a). In total, we profiled the transcriptome of 373 human MoDCs across 15 conditions (23–31 cells per condition; Supplementary Data 1). After removing 31 cells (~8 %) through stringent quality metrics (Supplementary Figure 6), 342 cells remained for downstream analyses (18–30 cells per condition, Supplementary Tables 1 and 2). Notably, we observed similar distributions of average $\log_{10}$-transformed read count per million (CPM) across all conditions. We detected an average of 10,820 genes (range: 9698–12,143) above an average 1 CPM in at least one experimental group and an average of 4221 genes (range: 3636–4827) below the 1 CPM average, respectively (Supplementary Figure 7A).

**Transcriptional reprogramming following *Salmonella* infection.** We applied the diffusion map non-linear dimensionality reduction method to reduce the high-dimensional normalized expression data set and to visualize relations between data points in a low-dimensional plot[18]. The resulting embedding highlights the progression of cells challenged with bacteria through markedly distinct stages, reflecting the sequential time points of the experiment. Notably, mock-infected cells displayed a shorter and continuous trajectory illustrating a more limited transcriptional drift in the absence of bacterial stimuli (Fig. 1b).

To identify transcriptomics changes taking place in MoDCs over the course of infection, we ordered all 342 cells in pseudotime using a set of 2,759 genes differentially expressed between *Salmonella*-challenged cells at 2 h and 6 h p.i. (*q*-value < 0.001) and identified genes that changed as a function of pseudotime across all cells (3188 genes, *q*-value < 0.001; Fig. 1c and Supplementary Data 2).

Genes partitioned into three broad co-regulated modules based on their single-cell expression profiles. Module 1 was strongly enriched for Gene Ontology (GO) terms related to the host immune response (*Innate Immune Response, Cytokine-mediated*

*signalling pathway*; Bonferroni-corrected *P*-value < 1e−39) and included genes known to be activated upon bacterial stimulation and internalization, such as pro-inflammatory cytokines (*IL12B, IL6, IL1B*) and the Lysosomal Associated Membrane Protein 3 (*LAMP3*) (Fig. 1c and Supplementary Data 3). In contrast, Module 2 and Module 3, which contained genes downregulated in challenged MoDCs over the course of the infection, were enriched for GO terms associated with metabolic activity (*Lipid Metabolic Process, Oxidation-Reduction Process*; Bonferroni-corrected *P*-value < 1e−03), and protein synthesis (*Ribosomal Biogenesis, Translational Initiation*; Bonferroni-corrected *P*-value < 1e−06), respectively (Fig. 1c and Supplementary Data 3). These findings are consistent with previous studies showing that professional phagocytes respond to bacterial stimuli by reorganizing their metabolic activities[19], in addition to mounting immune responses. Also, inhibition of host protein synthesis is a common strategy used by many viral and bacterial pathogens to disrupt host gene expression[20,21].

**Increasing MoDC heterogeneity over the course of infection.** As the marked temporal evolution of the transcriptional profile of *Salmonella*-challenged MoDCs masked the expectedly subtler differences expected in gene expression between bacterial strains (i.e., STM-LT2 vs. STM-D23580) and infection status (i.e., infected vs. bystander), we examined data collected at each time point in separate downstream analyses. To identify major cellular phenotypes, we identified unsupervised clusters at each time point by rank correlation between single-cell gene expression profiles using the *scran* package[22] (Supplementary Table 3).

At 2 h p.i. (Fig. 2), cluster 1 contained a balanced proportion of mock-uninfected and challenged MoDCs; cluster 3 was largely dominated by mock-uninfected cells and cluster 2 uniquely contained *Salmonella*-challenged MoDCs, possibly representing a small proportion of precocious activated MoDCs[23] (Supplementary Table 3). In agreement with this hypothesis, differential expression analysis (using the *scde* package[24]) revealed significant enrichment of genes involved in *Lipid Antigen Presentation* (Bonferroni-corrected *P*-value 5.94e-03; *CD1A, CD1B, CD1C*) as well as increased expression of surface markers (*CD83* and *CD40*) and cytokines (*TNF*), which are known markers of MoDC maturation, compared to all other cells at that time point (Fig. 2b and Supplementary Data 4 and 5).

At 4 h p.i. the different transcriptional profiles of uninfected, infected and bystander cells were more evident (Fig. 2). Cluster 3 contained all uninfected cells, with the exception of a single STM-D23580 bystander cell. Cluster 1 consisted of 74.5% of infected cells and cluster 2 was formed by 75.5% of bystander cells (Supplementary Table 3). Remarkably, at this stage of the infection, the comparison of unsupervised clusters 1 and 2 replicated a significant proportion of DE genes identified from the direct comparison of infected and bystander MoDCs in both STM-LT2 and STM-D23580 infections (Supplementary Figure 8A and Supplementary Data 4 and 6). Such findings indicate that clear distinctive transcriptional reprogramming events take place in cells that engulfed bacteria, relative to cells that were only exposed to extracellular cues. Conversely, the non-negligible proportion of cells from each experimental group split over multiple unsupervised clusters indicated a certain level of variability between cells exposed to uniform experimental stimuli.

Analysis of cluster 1 marker genes revealed a predominance of GO terms related to *Immune Response* and *Intracellular Stress* (Bonferroni-corrected *P*-value < 1e−06). In particular, this cluster was enriched for the lysosomal proteases *CTSB, CTSL, CTSS* suggesting an increased proteolytic activity that may occur in phagocytic cells harbouring bacteria. Cluster 2, containing most

of the bystander cells, was instead enriched in LPS-induced genes (Bonferroni-corrected *P*-value < 1.19e−11), compatible with extracellular stimuli. Cells in this cluster also express typical dendritic cells maturation markers (*CD40, CD83*) and increased levels of *TNF* and its targets (i.e. *TNFAIP2, TNFAIP3, TNFAIP8*) (Supplementary Data 4 and 5).

At 6 h p.i., cluster 3 was exclusively constituted by uninfected cells. Cluster 1 and cluster 2 revealed a more heterogeneous composition than that observed at 4 h p.i., emphasizing the complexity of cell-to cell variability (Fig. 2 and Supplementary Table 3). Genes induced in MoDCs of cluster 1 were enriched for *Antiviral Defence* factors, including interferon targets (Bonferroni-corrected *P*-value < 4.72e−11), whereas cells of cluster 2 were enriched for inflammatory genes and NF-kB targets (Bonferroni-corrected *P*-value < 4.14e−20) (Supplementary Data 5).

Interestingly, cluster 2 included most of the STM-D23580 bystander MoDCs (77.7%), in contrast to a minority of MoDCs infected with or exposed to STM-LT2 (31.8%), thus emphasizing larger differences between infected and bystander MoDCs challenged with STM-D23580 than those observed with STM-LT2 (Supplementary Table 3 and Supplementary Figure 8B). Consistent with this observation, we found 151 DE genes between infected and bystander MoDCs challenged with STM-D23580, compared to only 85 DE genes identified between infected and bystander MoDCs challenged with STM-LT2 at 6 h p.i. (Supplementary Figure 8C). Taken together, these results highlight more pronounced differences between invasive *Salmonella* infected cells and their equivalent bystander cells.

**STM-D23580 amplifies infected and bystander MoDC differences.** Next, we compared the differences between infected and bystander cells challenged with either *Salmonella* strain, to characterize the transcriptional reprogramming events common or specific to cells harbouring each of the two bacterial strains, in relation to their respective bystander cells.

ScRNA-seq revealed strain-specific changes and differential pathway activities in infected and bystander MoDCs. While certain genes were commonly up- or downregulated in infected and bystander cells by both bacterial strains, several genes were uniquely DE between infected and bystander cells only by STM-D23580 or STM-LT2. Indeed, these data suggest that the invasive strain achieves successful host infection by affecting common as well as disparate host pathways (Fig. 3 and Supplementary Data 7).

For instance, at 2 h p.i., STM-D23580-infected cells specifically upregulated the Membrane-Associated Ring-CH-Type Finger 8 *MARCH8*, which has been recently shown to be a target of the *Salmonella* effector protein SteD[25], as well as *RAB29*, a GTPase recruited to the *S.* Typhi-containing vacuole but not to vacuoles containing broad-host *Salmonella*[26].

Interestingly, at the later stages of our infection model, genes essential for correct antigen presentation (*CD83, CD40, HLA-DOA*) were downregulated exclusively in STM-D23580-infected cells relative to bystander cells.

**STM-D23580 enhances an inflammatory state in bystander MoDCs.** At 6 h p.i, our unsupervised clustering approach revealed an enrichment for STM-D23580 bystander MoDCs in cluster 2 (77.7%) compared to a more balanced representation of other cells across clusters 1 and 2 (Fig. 2 and Supplementary Table 3). These results suggest that most cells exposed to PAMPs from the invasive STM-D23580 engage in a distinct transcriptional program from that of STM-LT2 bystander MoDCs. As compared to STM-LT2, STM-D23580 bystander cells revealed a

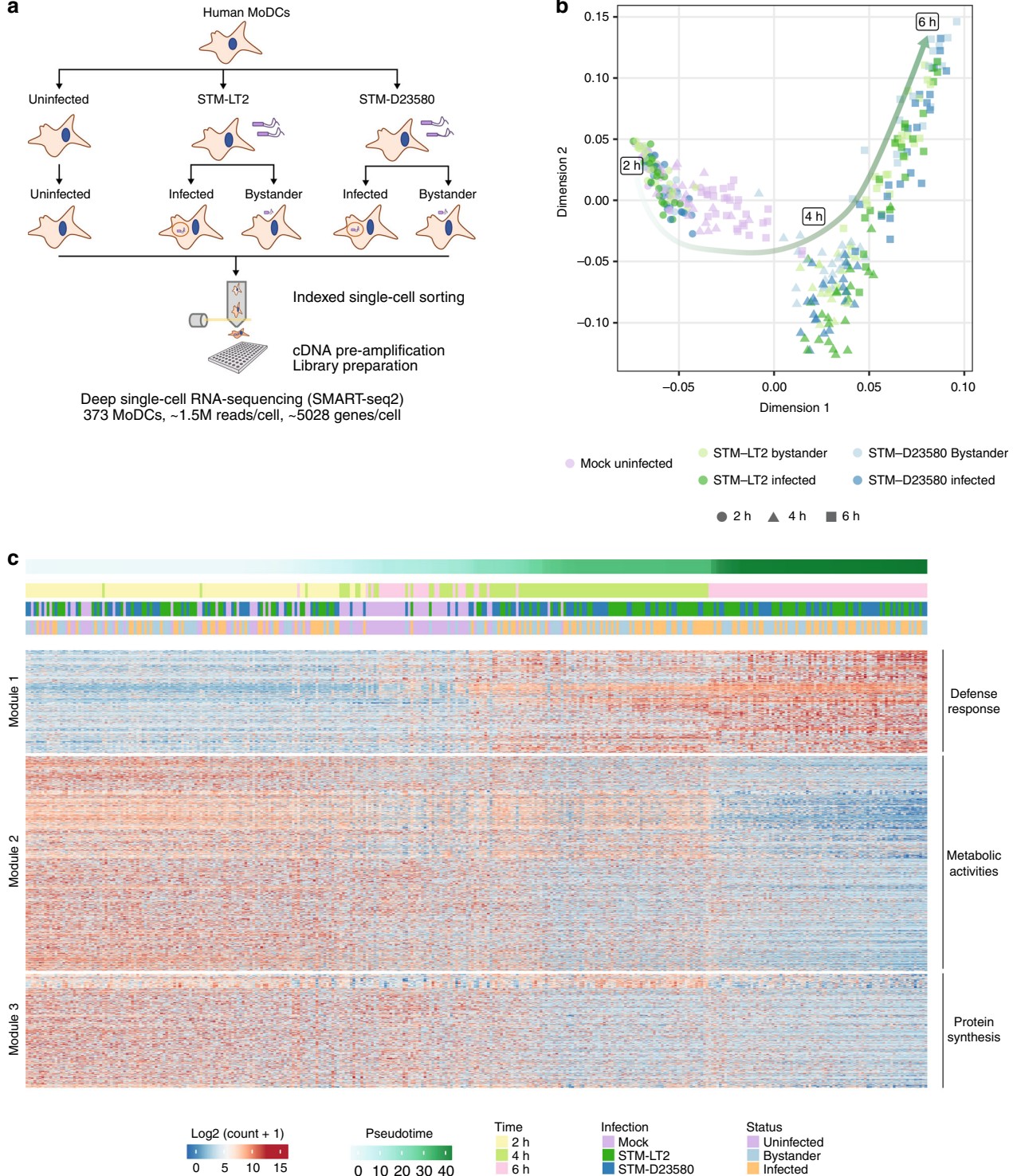

**Fig. 1** Single-cell transcriptomics analysis of human MoDCs challenged with invasive or non-invasive *Salmonella*. **a** Schematic representation of the experimental design. Human MoDCs were challenged with labelled STM-LT2 or STM-D23580 bacterial strains. At 2, 4 and 6 h post infection (p.i.), infected and bystander cells were sorted and processed according to the Smart-seq2 protocol. **b** The diffusion map non-linear dimensionality reduction technique was applied on the first 50 components of a principal component analysis (PCA) computed on the 500 most variable genes to visualize the high-dimensional relations between cells in the data set. The diffusion map reveals distinct cell populations that progress through the three time points of the experimental design. **c** Heat map displaying log₂-transformed normalized counts for 3188 genes that significantly change with pseudotime across all single-cells (*q*-value < 0.001). Pseudotime was computed from a set of 2759 genes differentially expressed between *Salmonella*-challenged cells at 2 and 6 h p.i. (*q*-value < 0.001). Experimental metadata (i.e., time point, infection, status) are indicated for each cell as colour bars above the heat map

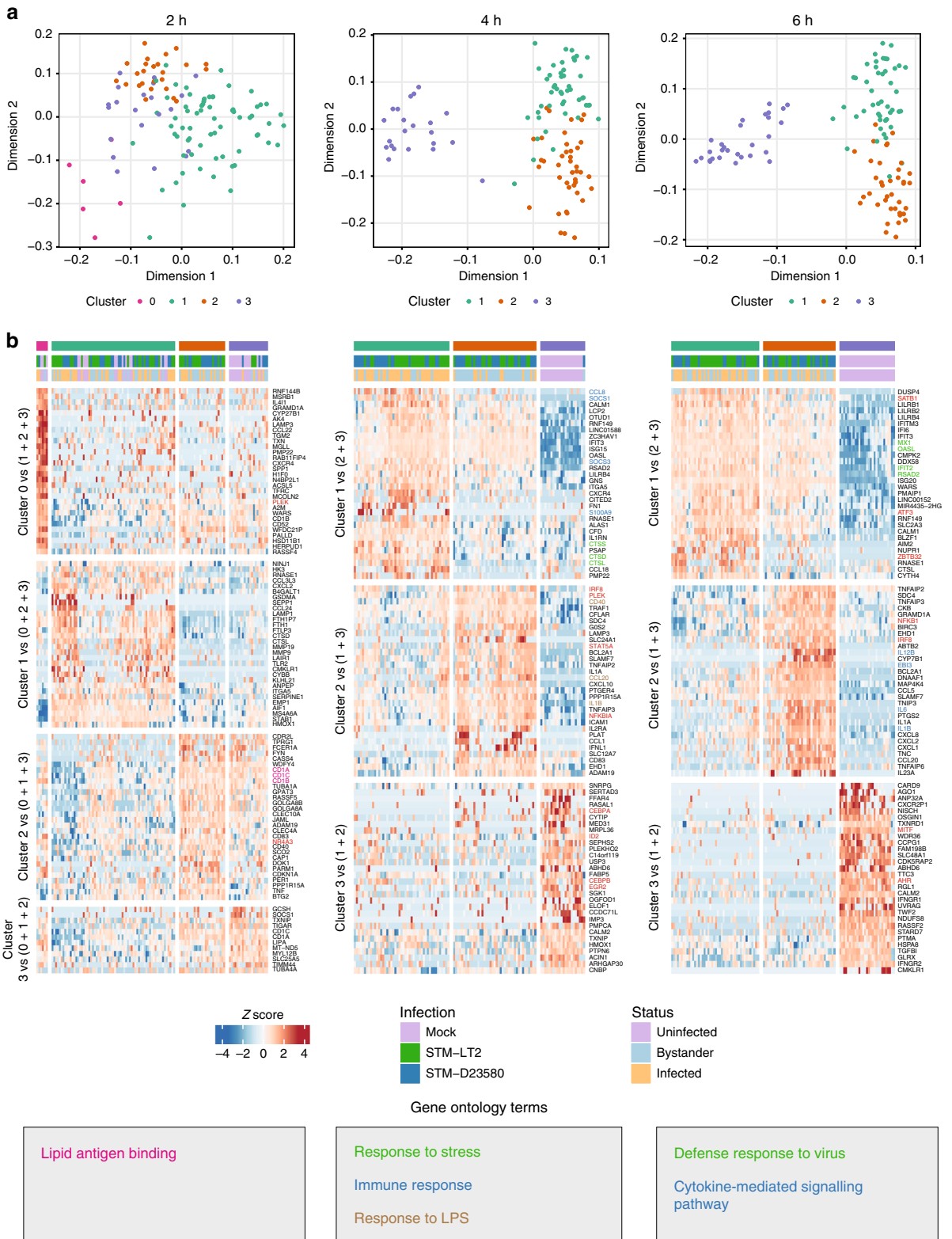

**Fig. 2** Unsupervised clustering reveals variability within infected and bystander cells. **a** Diffusion maps highlighting the unsupervised clusters identified by rank correlations between gene expression profiles of cells at each time point separately. Diffusion maps were computed on the first 50 components of a principal component analysis (PCA) using the 500 most variable genes identified within cells at each time point separately. **b** Heat maps displaying up to 30 significant marker genes identified as DE genes (P-value < 0.01) between each unsupervised cluster and all other cells at the corresponding time point (2, 4 and 6 h p.i, respectively). Rows and columns were clustered using Euclidian distance and complete linkage method applied to the row-scaled values across all heat map panels at that time point. Enriched GO categories and associated genes are indicated by colors. Transcription factors are highlighted in red

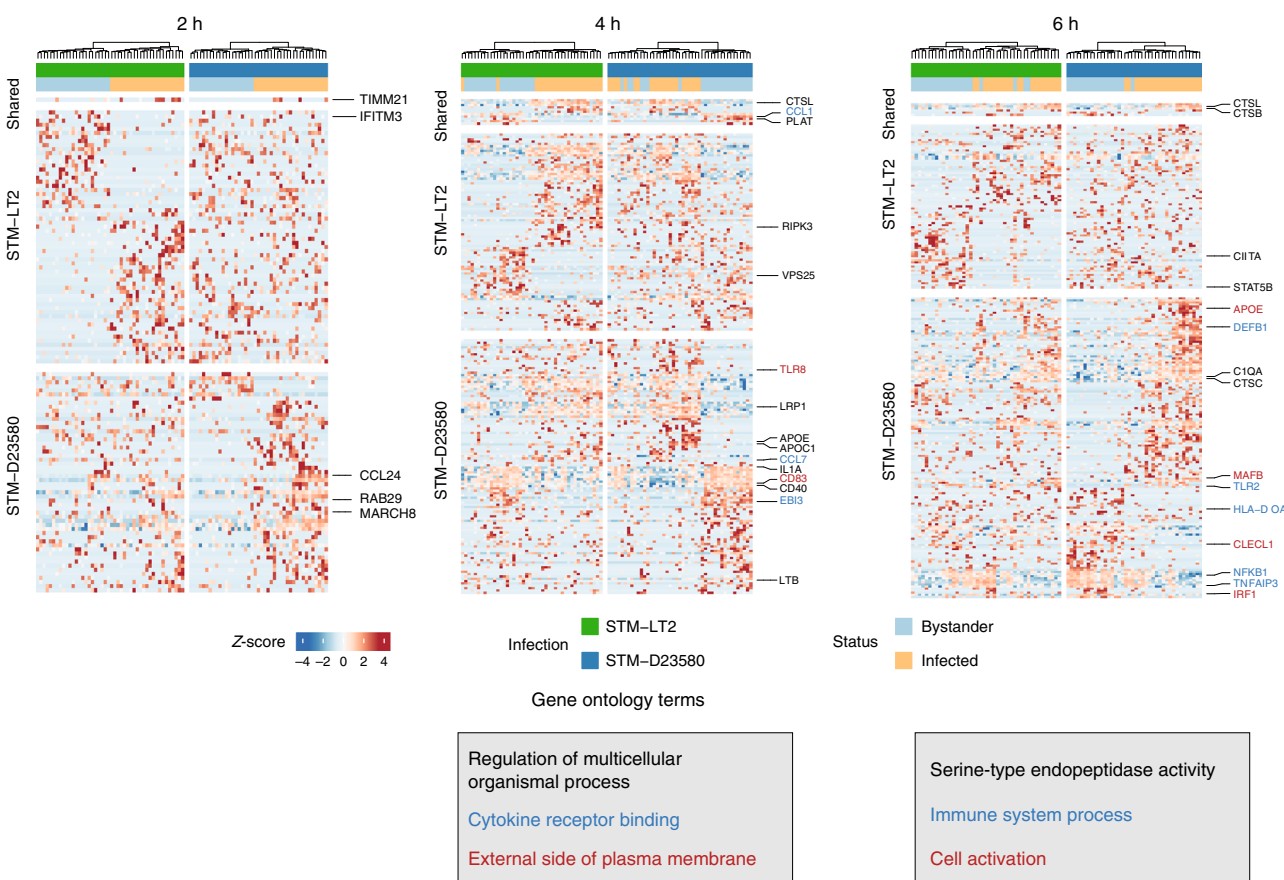

**Fig. 3** STM-D2350 and STM-LT2 induce divergent single-cell profiles in infected and bystander cells. Heat maps displaying significant DE genes (*P*-value < 0.01) identified between infected and bystander cells at time points 2, 4 and 6 h p.i, respectively. Rows and columns were clustered using Euclidian distance and Ward's clustering criterion applied to the row-scaled log-transformed normalized counts across all heat map panels at that time point. Enriched GO categories and associated genes are indicated by colors

significantly higher expression of the typical pro-inflammatory cytokines (*IL1B, IL12B, CXCL8*) (Fig. 4a and Supplementary Data 6 and 8). Additionally, the transcription factors *TNFAIP3, TNIP3, BIRC3*, which activate inflammatory signalling cascades, were upregulated in STM-D23580 bystander cells as compared to STM-LT2 (Fig. 4b and Supplementary Data 6 and 8). Exposure to invasive *Salmonella* also activated the transcription of genes known to inhibit apoptotic processes (*BCL2L1*) and regulate NF-kB signalling (*IKBKB*). Conversely, STM-D23580 bystander MoDCs expressed *ATF3*, a key negative regulator of cytokine expression in immune cells[27,28], at levels no different than uninfected cells. Such observation contrasts with a significant *ATF3* upregulation among STM-LT2 bystander cells. Together, these results demonstrate an impaired transcriptional regulation in cells exposed to the invasive strain. We next explored how STM-D23580 might direct enhanced pro-inflammatory response in exposed MoDCs. We assessed whether surface PAMPs might play a role by exposing MoDCs to heat-killed (HK) *Salmonella* treated with DNAse, RNAse and Proteinase K, and assayed pro-inflammatory cytokine responses. Higher IL-1β and IL-12p40 secretion was detected in MoDCs stimulated with HK STM-D23580 as compared to HK STM-LT2 (Supplementary Figure 9) suggesting that STM-D23580 surface moieties may play some part in this effect.

**Differences between STM-D23580- and STM-LT2-infected MoDCs.** To identify unique differences between MoDCs infected with either bacterial strain, we directly compared STM-

D23580- and STM-LT2-infected cells at each time point (Fig. 5a and Supplementary Data 6). During the early and intermediate stages of the infection (2 and 4 h p.i.), we found that STM-D23580-infected cells displayed increased expression of the apoptosis-related cysteine peptidase Caspase 3 (*CASP3*), the vacuolar protein sorting 25 (*VPS25*) required for multivesicular body (MVB) formation, and a group of genes involved in the metabolism of fatty acids (*MUT, CHST13, APOBEC3A*). Conversely, genes that facilitate ubiquitination and proteasomal degradation (*KLHL2, TRIM13*) were downregulated in STM-D23580 as compared with STM-LT2-infected cells. Consistent with progressive bacterial adaptation to the host intracellular environment, we found increased gene expression changes later in infection (6 h p.i.). Interestingly, the RING Type E3-ubiquitin Transferases *MARCH1* and *RNF139*, required for ubiquitination of MHC-II and MHC-I molecules[29,30] respectively, were upregulated in STM-D23580 as compared to STM-LT2-infected MoDCs (Fig. 5b and Supplementary Data 6 and 8).

We also found that the transferrin receptor *TFRC* was upregulated in STM-D23580-infected cells compared to both uninfected and STM-LT2 infected. This correlates with increased expression of *DNAJC13*, a protein that promotes recycling of transferrin directly from early endosome to plasma membrane[31] (Fig. 5b and Supplementary Data 6). Since iron is a critical nutrient resource that influences intracellular bacterial growth and virulence factor expression, these data suggest that invasive *Salmonella* usurps iron intake mechanisms to facilitate its intracellular survival and replication[32,33].

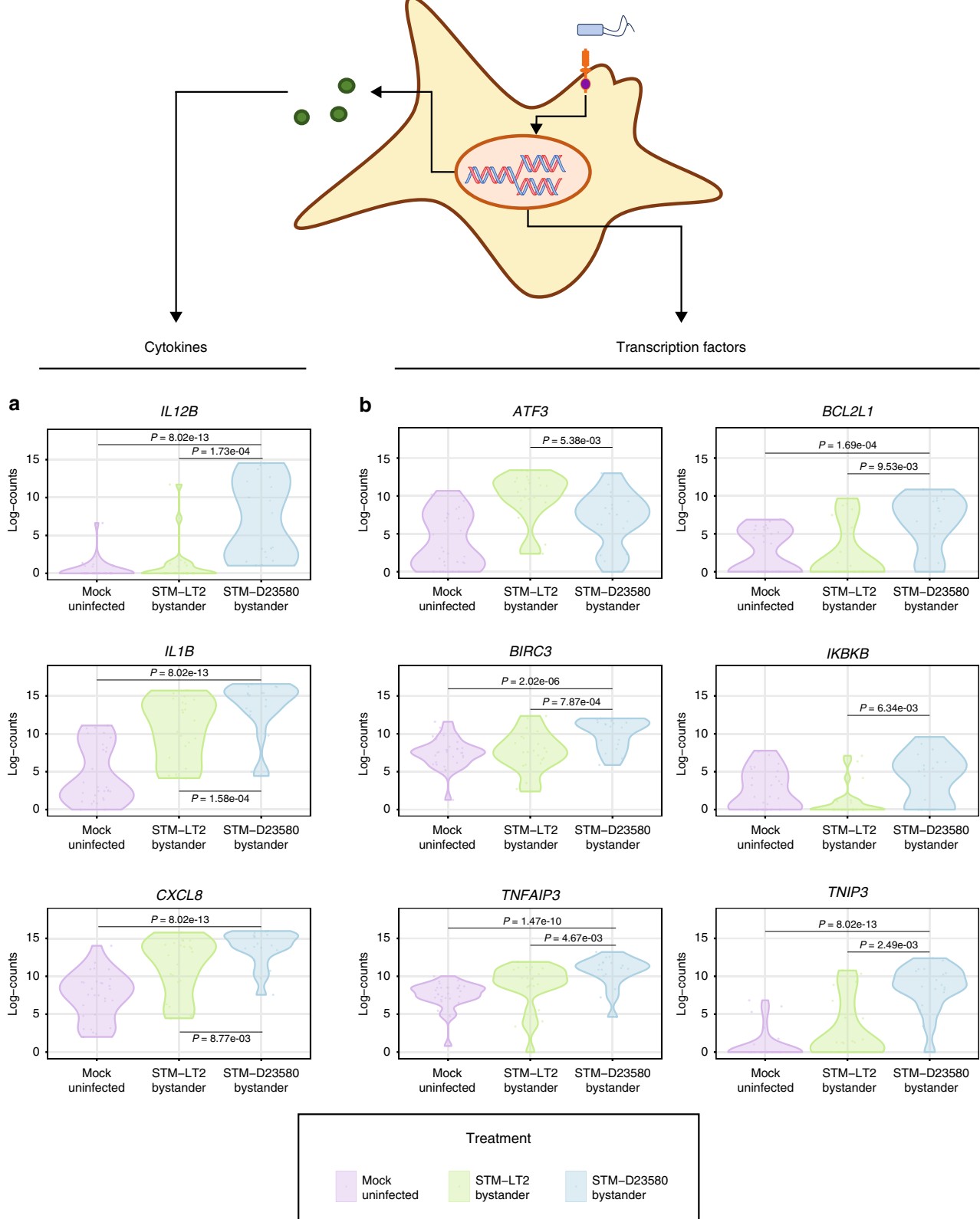

**Fig. 4** STM-D23580 bystander MoDCs show an exaggerated pro-inflammatory response. **a** Pro-inflammatory cytokines *IL12B*[(*)], *IL1B*[(*)] and *CXCL8*[(*)] and **b** transcription factors *BCL2L1*[(*)], *BIRC3*[(*)], *IKBKB*[(**)], *TNIP3*[(*)] and *TNFAIP3*[(*)] show significant (*P*-value < 0.01) upregulation in STM-D23580 bystander cells compared to both STM-LT2 bystander MoDCs and uninfected control[(*)] or compared to STM-LT2 bystander MoDCs only[(**)]. *ATF3* shows significant downregulation in STM-D23580 bystander cells compared to STM-LT2 bystander MoDCs

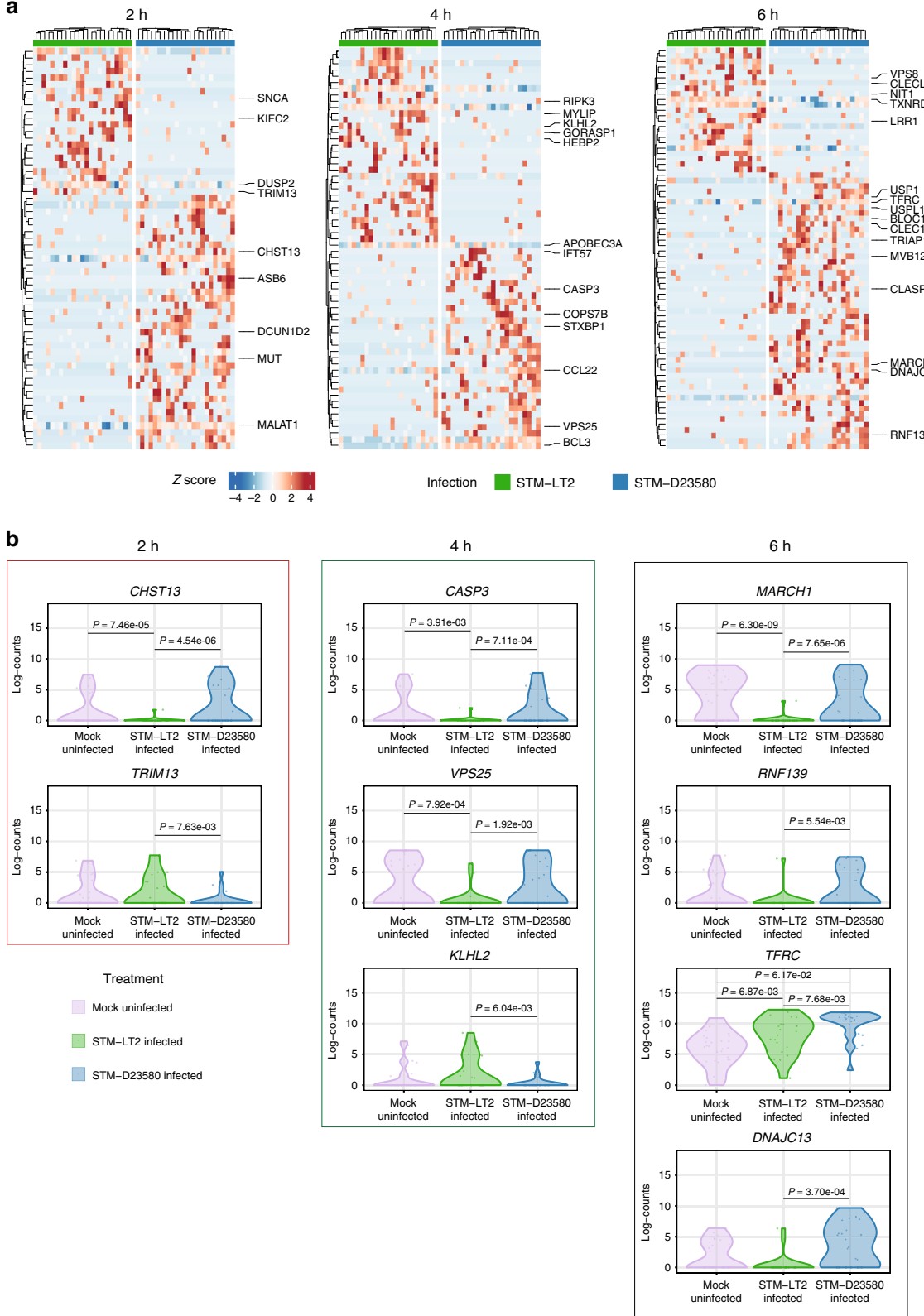

**Fig. 5** Direct comparison of MoDCs infected with STM-D23580 or STM-LT2. **a** Heat maps displaying significant DE genes (*P*-value < 0.01) identified between STM-D23580 and STM-LT2-infected cells at each time point (2, 4 and 6 h p.i, respectively). Rows and columns were clustered using Euclidian distance and complete linkage method applied to the row-scaled values across all heat map panels at that time point. **b** Violin plots displaying the log-transformed normalized gene expression level of relevant DE genes (*P*-value < 0.01) identified between STM-D23580- and STM-LT2-infected MoDCs

**STM-D23580 exploits the *IL10/MARCH1/CD83* axis**. Intracellular *Salmonella* can subvert MoDC functions by manipulating cell-intrinsic host immune pathways[34,35]. Consistent with this notion, we discovered that cells infected with STM-D23580 expressed a significantly higher level of *IL10* and *JAK2* relative to uninfected MoDCs at 6 h p.i. (Fig. 6a). We also found that *CLECL1*, involved in T cell activation and co-stimulation[36], and *CLEC7A*, implicated in pattern recognition of pathogens leading to the activation of pro-inflammatory cytokines[37], were both significantly downregulated at 6 h p.i. in STM-D23580-infected cells compared to both unstimulated MoDCs and STM-LT2-infected MoDCs (Fig. 6a). These data indicate that a suppressive phenotype develops in STM-D23580-infected MoDCs at later stages of infection.

MoDC maturation is also marked by the induction of the surface activation marker CD83, in addition to MHC-II and CD86. No significant differences were observed in *CD83* expression among infected and bystander STM-LT2 challenged MoDCs. However, MoDCs infected with invasive *Salmonella* showed a bimodal expression of CD83, with a subset of cells failing to upregulate this receptor, both at the transcriptional and the protein level (Fig. 6b, c). Using flow cytometry, we confirmed a statistically significant reduction of CD83 surface expression in STM-D23580 relative to STM-LT2-infected MoDCs (Fig. 6d). MARCH1 is an E3-ubiquitin ligase which targets MHC-II and CD86 for degradation[38,39]. Upon MoDC activation, upregulation of CD83 is associated with a decrease in *MARCH1* mRNA expression[40]. Other studies demonstrated that the anti-inflammatory cytokine IL-10 induces MARCH1[41,42]. We detected significantly higher amounts of IL-10 in the supernatants of MoDCs challenged with STM-D23580 as compared to STM-LT2 at later stage of the infection (Supplementary Figure 10). This correlated with higher expression of *MARCH1* in STM-D23580-infected cells compared to STM-LT2 infected (Fig. 7a). The lack of downregulation of *MARCH1* in STM-D23580-infected cells was confirmed by qPCR (Fig. 7b).

To investigate whether intracellular STM-D23580 could affect the antigen presentation machinery by acting through the *IL10/CD83/MARCH1* axis, we analysed surface expression of HLA-DR and CD86 on infected MoDCs by FACS. We observed that intracellular STM-D23580 correlated with a significant reduction in surface HLA-DR and CD86 expression compared with STM-LT2 (Fig. 7c). In contrast, there was no difference in the surface expression of HLA-I, a known target of MARCH9[43] and CD1a, which traffics independently of MHC-II during MoDC maturation[44]. These results suggest that invasive *Salmonella* develops molecular mechanisms to specifically target MHC-II molecules (Supplementary Figure 11).

Interestingly, we detected decreased expression of CD1a, necessary to present lipid antigens[45], at the cell surface on infected cells compared to unstimulated and bystander cells (Supplementary Figure 12A), consistent with our single-cell data (Supplementary Figure 12B). On the contrary HLA-DR and HLA-I surface expression were upregulated in infected MoDCs compared to both bystander or uninfected cells.

**STM-D23580 impairs antigen presentation to CD4$^+$ T cells**. Considering the role of CD83 in T cell stimulation[46], we tested whether reduced CD83 surface expression on infected MoDCs correlated with diminished T cell stimulatory capacity. We developed an in vitro antigen presentation assay where *Salmonella* infected MoDCs were co-cultured with CD4$^+$ T cell clones specific for PhoN[47], a protein expressed in multiple *Salmonella* serovars, including *S*. Typhimurium. At 6 h p.i., we FACS sorted infected cells based on their CD83 surface expression. While

STM-LT2 infected MoDCs were all CD83$^+$ (Fig. 8a and Supplementary Figure 13A), STM-D23580-infected MoDCs could be separated into CD83$^{-/low}$ and CD83$^{high}$ cells (Fig. 8b and Supplementary Figure 13B). Uninfected MoDCs were used as CD83$^{-/low}$ control (Fig. 8c and Supplementary Figure 13D). We then co-cultured the sorted MoDCs with PhoN-specific T cell clones and tested them for their ability to secrete IFN-γ and TNF-α, by intracellular cytokine staining. STM-LT2 and STM-D23580 CD83$^{high}$ infected MoDCs triggered CD4$^+$ T cell activation, with production of TNF-α and IFN-γ. In contrast, T cells co-cultured with STM-D23580 CD83$^{-/low}$ infected MoDCs produced significantly lower levels of these cytokines (Fig. 8d,e and Supplementary Figure 13E).

We corroborated the specificity of our assay using CD4$^+$ T cell clones recognizing HlyE, a protein expressed in *S*. Typhi but not *S*. Typhimurium[47,48]. As expected, HlyE-specific CD4$^+$ T cell clones responded exclusively to CD83$^+$ ST-Ty2-infected MoDCs (Supplementary Figure 13C and 13E). In addition to cytokine production, our co-culture model indicates that invasive *Salmonella* may also affect T cell costimulatory signals, such as the engagement of CD40 with CD40L. In comparison to CD83$^{high}$ STM-D23580-infected cells, we observed a reduction of CD40L expression among cytokine-producing T cell clones co-cultured with CD83$^{-/low}$ STM-D23580-infected MoDCs (Fig. 8f, g), suggesting that the suboptimal activation of T cells in the presence of invasive *Salmonella* might influence also the expression of costimulatory molecules on T cells. Together, these data indicate that the intracellular activity of invasive *Salmonella*, in combination with reduced CD83, may impair the ability of MoDC subsets to trigger antigen-specific T cell activation.

**Small bulks recapitulate average RNA-sequencing profiles**. To validate our scRNA-seq findings, we conducted bulk RNA-seq of 5000 cells from infected and bystander MoDCs treated with either invasive or non-invasive *Salmonella* using a modified version of the Smart-seq2 protocol. Principal component analysis (PCA) of these bulk samples revealed a distribution of the various experimental conditions comparable to the diffusion map representation of our single-cell data. These results confirm the distinction between uninfected and challenged MoDCs, but also bystander and infected cells (Supplementary Figure 14A). Differential expression analysis of bulk samples confirmed higher expression of *CTSD, LGMN, CCL7, CCL26, MS4A4A, LAMP1* and *APOE* in infected cells, and *IL1B, PLAT, CD1C, CD1A, CCL1, TRAF4* and *CD83* in bystander cells (Supplementary Figure 14B and Supplementary Data 9). We also confirmed that *MARCH1* expression in STM-D23580-infected cells was comparable to uninfected cells, which contrasted with a marked downregulation of *MARCH1* expression in STM-LT2-infected MoDCs. In addition, we confirmed that *CD83* and *IL10* expression was differentially regulated between STM-LT2- and STM-D23580-infected cells, although the bulk method failed to resolve the cell-to-cell variability detected only with scRNA-seq (Supplementary Figure 14C).

**Independent validation using qPCR**. We validated key genes identified as differentially expressed between infected and bystander MoDCs using qPCR. First, we identified candidate markers from genes identified both in our previous analyses of unsupervised clusters and direct comparisons (i.e., infected relative to bystander MoDCs or STM-D23580 relative to STM-LT2). After FACS sorting and cDNA preparation according to the Smart-seq2 protocol, we then pooled cDNA from individual cells from each group and performed qPCR for selected genes. Statistical analysis of the relationship between our qPCR and

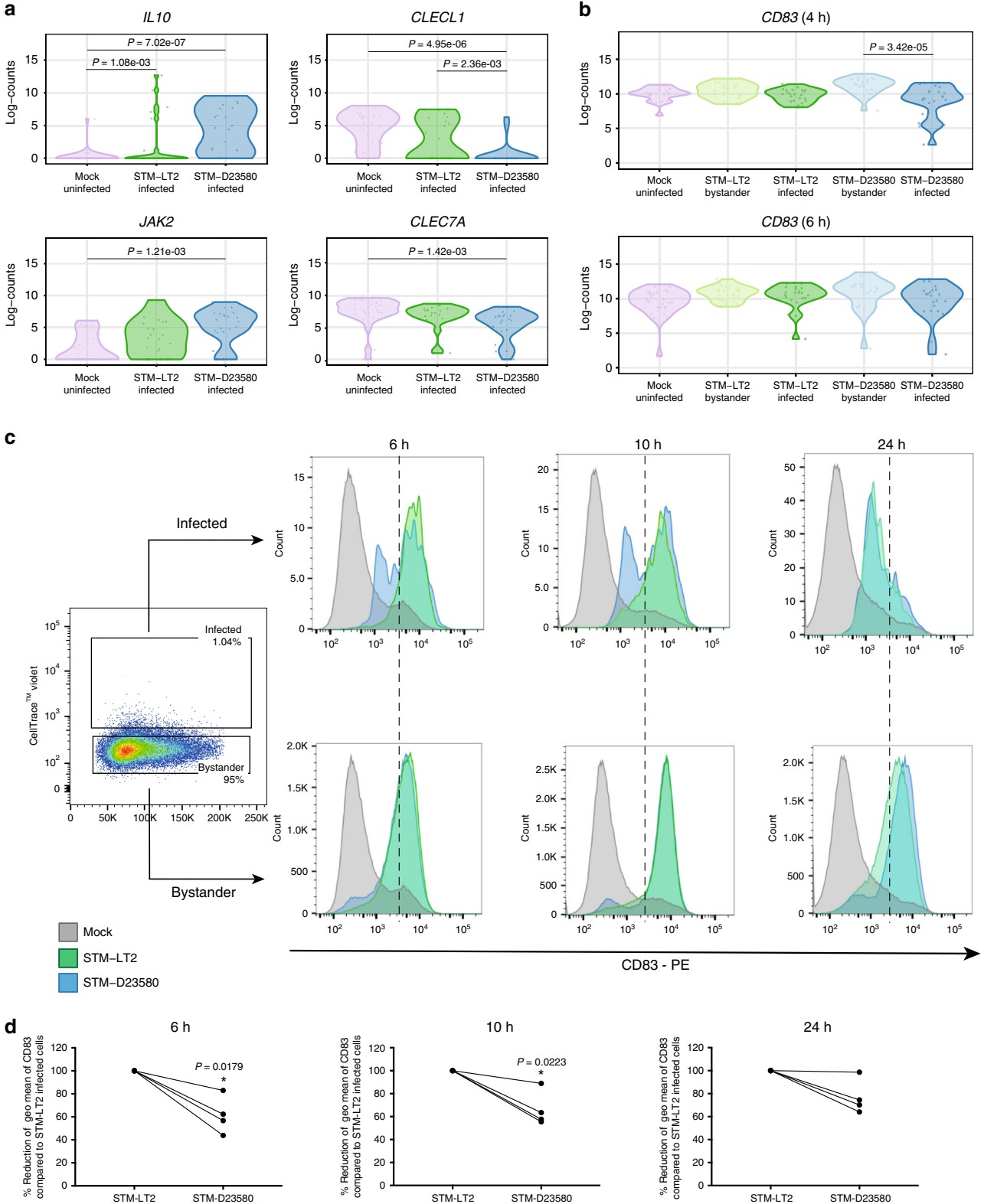

**Fig. 6** Intracellular STM-D23580 reduces CD83 surface expression. **a** Violin plots showing single-cell gene expression of *IL10, JAK2, CLECL1* and *CLEC7A* at 6 h p.i. in infected and uninfected cells. **b** Violin plots showing single-cell gene expression of *CD83* at 4 and 6 h p.i. in all experimental groups. **c** Flow cytometry histograms showing the expression of surface CD83 on gated infected and bystander cells challenged with STM-D23580 (blue), STM-LT2 (green) or left uninfected (grey) for 6, 10 or 24 h. STM-D23580-infected MoDCs display decreased surface expression of CD83 as compared with STM-LT2. Histograms from one representative example of at least four biological replicates. **d** STM-D23580 infection reduces surface CD83 levels detected by flow cytometry. On the *y*-axis, the geometric mean of fluorescence levels for CD83 on STM-D23580-infected cells is represented as a percentage of the level in STM-LT2-infected cells, set at 100%. Four independent experiments are shown. Two-tailed paired Student's *t*-test, *P*-value < 0.05 (*)

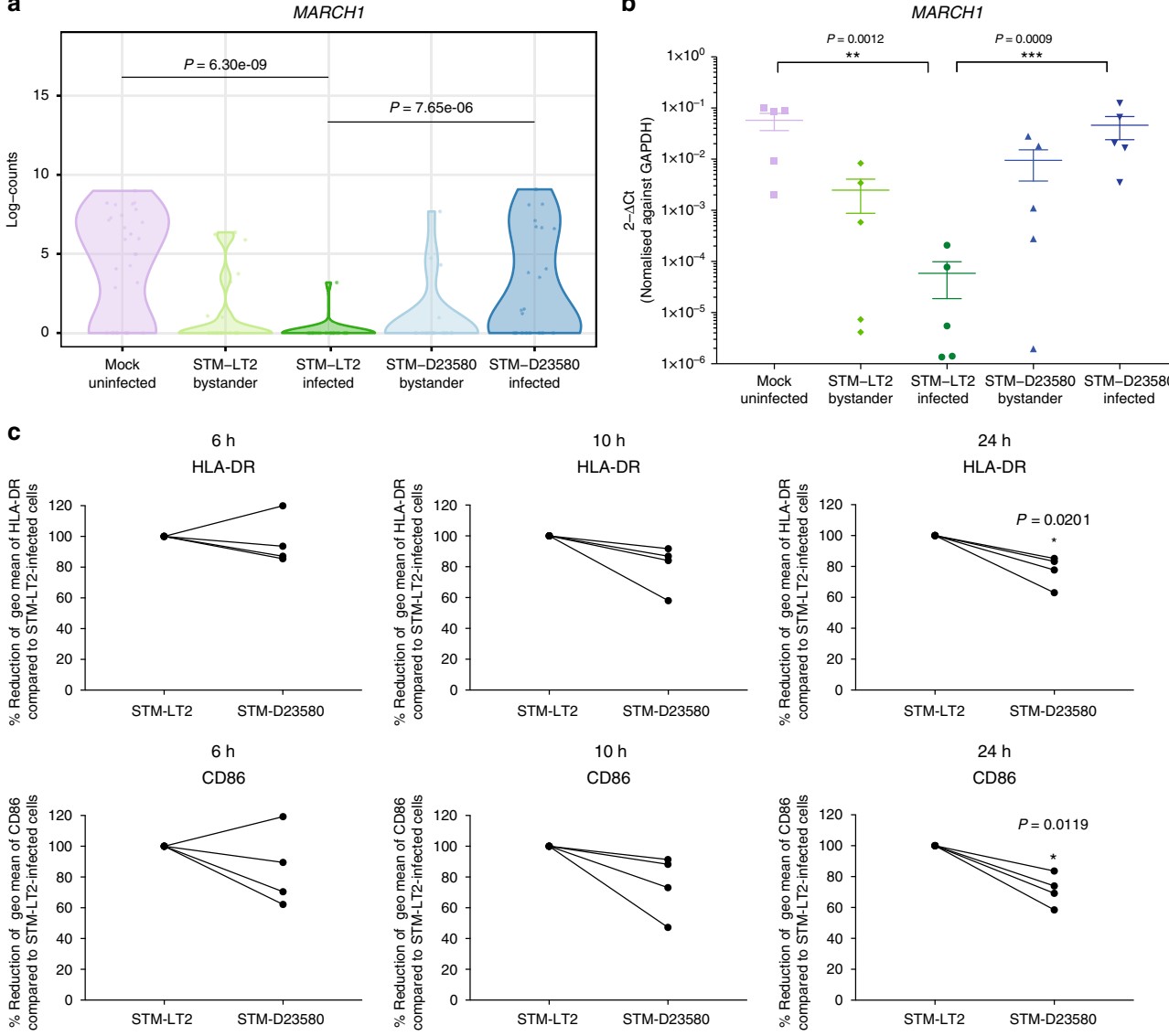

**Fig. 7** Intracellular STM-D23580 reduces HLA-DR and CD86 surface expression compared to STM-LT2. **a** Violin plot showing single-cell gene expression of *MARCH1* at 6 h p.i. in all the experimental groups. **b** Dot plot showing gene expression of *MARCH1* at 6 h p.i detected by qPCR in all experimental groups. The mean±SEM of five independent experiments is shown. Two-way ANOVA test, *P*-value < 0.01(**), <0.001(***). **c** Surface HLA-DR and CD86 expression measured by flow cytometry. On the *y*-axis, the geometric mean of fluorescence levels for HLA-DR and CD86 on STM-D23580-infected cells is represented as a percentage of the level in STM-LT2 infected cells, set at 100%. Four independent experiments are shown. Two-tail paired Student's *t*-test, *P*-value < 0.05 (*)

scRNA-seq results revealed a significant correlation between the two methods across all genes tested (Supplementary Figure 15). This analysis confirmed that infected cells induced *CTSL, CTSD* and *MSA4A4A*, while bystander cells predominantly expressed *PLAT, EBI3* and *STAT5B*. We also validated *MARCH1* as a marker of STM-D23580-infected MoDCs and *VPS25* as a marker for the STM-LT2 infected population (Supplementary Figure 15).

## Discussion
Here we define how invasive *Salmonella* STM-D23580 directs divergent and heightened heterogeneity of single-cell profiles of infected and bystander MoDCs compared to non-invasive STM-LT2. Specifically, invasive *Salmonella*-containing MoDCs differentially regulated genes associated with endosomal trafficking and antigen presentation pathways. Invasive *Salmonella* also induced *IL10* and downregulated *CD83* likely via its inability to

downregulate *MARCH1*. These mechanisms evoke suboptimal signals for efficient CD4$^+$ T cell activation. MoDCs exposed to, but not harbouring replication-competent invasive *Salmonella*, expressed increased levels of pro-inflammatory transcription factors and cytokines. We then validated these results by both re-sequencing sorted subsets, and at mRNA and protein levels in independent donors. In summary, our scRNA-seq analyses reported here reveal hyper-activation of bystander MoDCs and mechanisms of cell-intrinsic host adaption exploited by *Salmonella* ST313 with implications for its invasive behaviour in immunocompromised hosts.

Heightened activation of bystander MoDCs following challenge with invasive *Salmonella* may facilitate recruitment of adaptive immune cells to non-infected rather than infected MoDCs and serve a putative decoy role. Divergent responses in bystander MoDCs may arise in part from differences in the PAMP composition of the STM-D23580 surface coat, since treatment with of

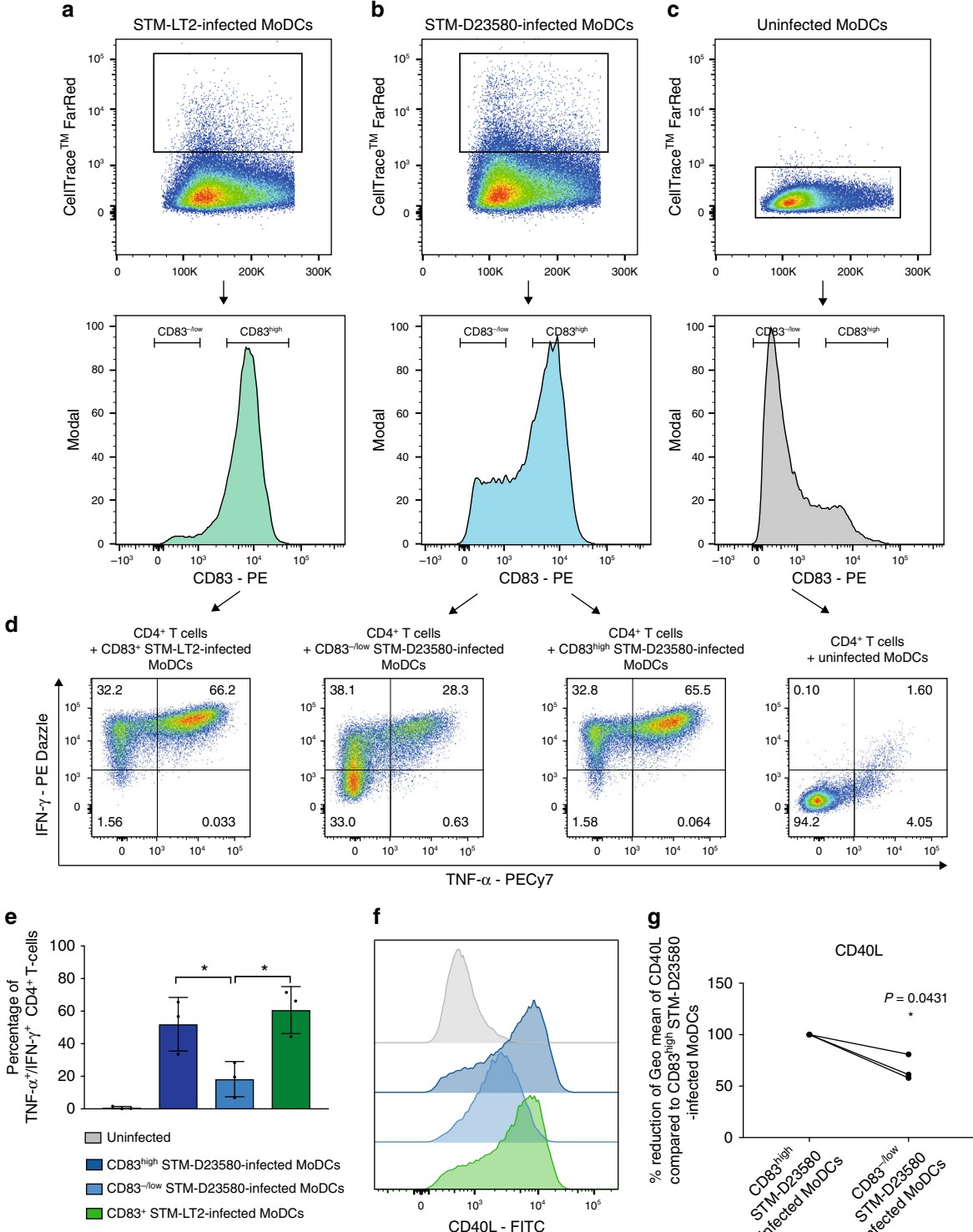

**Fig. 8** CD83[−/low] STM-D23580-infected MoDCs direct defective activation of *Salmonella*-specific CD4[+] T cells. At 6 h p.i., **a** STM-LT2-infected MoDCs were sorted as CD83[+] while **b** STM-D23580-infected MoDCs were sorted as CD83[−/low] or CD83[high]. **c** Uninfected MoDCs were used as negative control. Sorted subsets of infected MoDCs were co-cultured overnight with PhoN-specific CD4[+] T cell clones cross-reactive against typhoidal and non-typhoidal serovars. **d** Capacity of CD4[+] T cell to secrete IFN-γ and TNF-α, as detected by intracellular staining. FACS plots show the percentage of IFN-γ[+]/TNF-α[+] cells out of total CD3[+] CD4[+] T cells. Dot plots comprise a representative experiment out of four. **e** CD83[−/low] STM-D23580-infected MoDCs induce a significantly lower amount of cytokine-producing T cells, as compared to CD83[high] STM-D23580 or CD83[+] STM-LT2 infected cells. The mean±SEM of three independent experiments is shown One-way ANOVA test, *P*-value < 0.05 (*). **f** Flow cytometry histogram showing CD40L expression on T cells co-cultured with CD83[+] STM-LT2 infected MoDCs, CD83[−/low] or CD83[high] STM-D23580-infected cells. Histograms from one representative example of at least three biological replicates. **g** Geometric mean of fluorescence intensity for CD40L in T cells co-cultured with CD83[−/low] STM-D23580-infected MoDCs, represented as a percentage of reduction from CD83[high] STM-D23580-infected MoDCs. three independent experiments are shown. Two-tailed paired Student's *t*-test, *P*-value < 0.05 (*)

DNAase, RNAase and proteinase K did not impair its immuno-stimulatory ability. We also observed widespread changes in expression levels of key transcription factors in MoDCs exposed to STM-D23580 with corresponding induction of pro-inflammatory cytokines, such as IL-12p40 and IL-1β.

Cells that internalized STM-D23580 displayed reduced maturation and downregulated components of the antigen presentation machinery, likely via induction of *IL10* and dysregulated expression of *MARCH1*. *IL10* induction as an evasion strategy by STM-D23580 is in line with previous observations that explain how increased IL-10 expression enhances susceptibility to invasive NTS infection and blunts intestinal inflammation[49]. Consistent with inducing IL-10, both at transcriptional and protein level, STM-D23580 also induced *MARCH1*. MARCH1 is an E3 ubiquitin ligase that plays an important role in ubiquitination and degradation of MHC-II and CD86[38,39,50]. In particular, the enzymatic function of MARCH1 towards MHC-II and CD86 only occurs in the absence of appreciable levels of CD83[40], an immunoglobulin-like surface molecule that normally blocks MHC-II/CD86 association with MARCH1. Differential expression of the *IL10/CD83/MARCH1* axis by invasive *Salmonella* represents a coordinated strategy employed by this strain to evade effective antigen presentation in cells harbouring live bacteria. To our knowledge, this is the first description of such an evasion mechanism by this pathovar.

We propose that invasive *Salmonella* impairs the ability of cells to present antigen to CD4+ restricted T cells by reducing surface MHC-II. Our proposal agrees with studies showing that infection with *Salmonella* inhibits antigen presentation by murine MoDCs[51]. Here, we employed PhoN-specific CD4+ T cells clones generated from volunteers diagnosed with enteric fever[47]. We observed that the subset of STM-D23580-infected MoDCs with CD83−/low expression showed significant defects in their ability to elicit production of TNF-α and IFN-γ by CD4+ T cells. The defective T cell costimulatory capacity of CD83-/low STM-D23580-infected MoDCs is consistent with the observation that specific siRNA-mediated knockdown of CD83 has a marked effect on the capacity of MoDCs to stimulate T cell responses[46]. MoDCs infected with the Opa$_{CEA}$-expressing strain of *Neisseria gonorrhoeae* did not display any surface expression of CD83 and showed a reduced capacity to stimulate an allogeneic T cell proliferative response[52]. Other pathogens that establish latency, such as herpes simplex virus type 1 (HSV-1), human cytomegalovirus (HCMV) and vaccinia virus, adopt different strategies to interfere with CD83 expression on the surface of infected MoDCs. This process prevents MoDC-mediated T cell activation and proliferation[53–56]. For example, HCMV causes either CD83 degradation in a proteasome-dependent manner or shedding from the surface of mature MoDCs[56,57]. HSV-1 induces both mRNA and protein degradation of CD83 in mature MoDCs[53,55,58]. Intriguingly, surface expression of CD83 appears downregulated only by the ST313 *Salmonella* strain, while both STM-LT2 and the typhoidal strain Ty2 show comparable CD83 levels.

A positive regulator like CD40L when associated with CD40 not only activates T cells, but also activates MoDCs. Pathogens can downregulate CD40 expression on APCs. For instance, *Helicobacter pylori* reduces the expression of CD40L on T cells, exploiting the CD40/CD40L pathway for its survival[59]. *Bordetella pertussis* and *Bordetella bronchiseptica* both decrease surface levels of CD40 on MoDCs to promote development of regulatory T cells[60,61]. A significant role for CD40/CD40L interactions exists to mount a successful cellular immune response against *Salmonella* in normal BALB/c mice[62]. In addition, MHC-II, CD80, CD86 and CD40 are downregulated during chronic infection with BCG[63] and both in in vitro and lepromatous patients by

*Mycobacterium leprae*[64,65]. Co-stimulation not only amplifies the magnitude of T cell and APC interaction, but also fine-tunes the downstream immune signalling.

Studies using scRNA-seq to map host responses to *Salmonella* in murine macrophages revealed that variable PhoP/PhoQ activity leading to LPS modification underlies this heterogeneity in host response[66]. Macrophages can induce either alternative pro-inflammatory M1-like or anti-inflammatory M2-like maturation states, depending on the replication status of *Salmonella*[16]. So far, studies identified three subpopulations of infected macrophages corresponding to consecutive stages of infection[67]. Our work extends this heterogeneity in cell-to-cell response to human MoDCs. Our results also demonstrate that scRNA-seq can uncover subtle differences in host-pathogen interactions between genetically similar *Salmonella* strains.

A recent study demonstrated that CD11b+ migratory DCs facilitate STM-D23580 hyper-dissemination to systemic sites in vivo[68]. The scRNA-seq analysis presented here elucidates fine mechanisms by which the invasive STM-D23580 strain manipulates gene expression of infected and bystander MoDCs. We propose that downregulation of costimulatory pathways by invasive *Salmonella* results in a response detrimental to the host. Since any negative effect on antigen presentation can further accentuate inherent CD4+ T cell defects observed in immunocompromised individuals, our results delineate a possible molecular pathway exploited by ST313 to evade immediate detection. Overall, these observations contribute to a better understanding of the pathogenesis and dissemination of invasive NTS disease.

## Methods

**Bacterial strains and labelling.** *Salmonella enterica* serovar Typhimurium (STM) strain LT2 (ATCC 700220) and the clinical isolate STM-D23580, were used as representative non-typhoidal *Salmonella* sequence type 19 (ST19) and type 313 (ST313), respectively. Strain LT2 is one of the principal *Salmonella* laboratory strains used in cellular and molecular biology[69]. Strain D23580 was isolated from the blood of an HIV-negative Malawian child with malaria and anaemia. *Salmonella enterica* serovar Typhi (ST) strain Ty2 (ATCC 700931) was used as representative typhoidal serovar. All bacteria were grown to logarithmic growth phase in LB Lennox broth (Sigma) supplemented with sucrose (Sigma) at a final concentration of 10%. Aliquots were kept frozen at –80°C until use, while bacterial viability was monitored periodically. For each experiment, an aliquot of bacteria was thawed, diluted in RPMI 1640 (Sigma) and incubated in the presence of 5uM of CellTrace™ Violet or CellTrace™ Far Red Cell Proliferation kit (Thermo-Fisher) at 37°C for 20 min while shaking (200 rpm). Bacteria were then washed and re-suspended in RPMI to obtain a multiplicity of infection (MOI) of 10:1. The number of microorganisms was assessed at each time point post infection by plating 10-fold dilutions of the bacterial suspension, in quadruplicate, on LB Lennox agar (Sigma). The number of bacteria was determined as colony forming units (CFU).

**Preparation of heat-killed bacteria.** *Salmonella* strains were grown in LB broth supplemented with 10% sucrose. Bacteria were harvested during the logarithmic growth phase, diluted to an OD 0.5, and killed by heating at 100 °C for 15 min. The suspension of heat-killed (HK) bacteria was allowed to cool at room temperature (RT) for 15 min and, subsequently, plated on LB agar to confirm bacteria were no longer viable. Enzymatic digestion of nucleic acid was performed using 10 mg/mL of DNase I and RNase (Qiagen) for 30 min at 37°C, followed by 10 mg/mL of Proteinase K (Qiagen) at 59°C for 3 h. Decreasing amount (50–1 μL) of the HK bacterial suspensions were used to stimulate MoDCs. LPS from *S. enterica* serovar Typhimurium (Sigma) was used as a control (100–0.01 ng/mL).

**Generation and infection of MoDCs.** Leukocyte Reduction System cones were obtained from the UK National Blood Centre with informed consent following local ethical guidelines. Blood was diluted in phosphate-buffered saline (PBS) and layered on a standard density gradient (Lymphoprep™). Peripheral Blood Mononuclear Cells (PBMC) were collected from the interface and washed in PBS at 4 °C. Monocytes were obtained by magnetic positive selection, using the human anti-CD14+ MicroBeads (Miltenyi Biotec, Germany) according to the manufacturer's protocol.

Freshly isolated monocytes were plated in tissue culture treated dishes (Falcon) at a density of 1E + 06/mL. Differentiation into monocyte-derived dendritic cells (MoDCs) was induced in the presence of 40 ng/mL of recombinant human (rh)

granulocyte macrophage-colony-stimulating factor (GM-CSF; PeproTech) and 40 ng/mL rh Interleukin-4 (IL-4; PeproTech).

After 5 days of culture, MoDCs were harvested, re-suspended in RPMI supplemented with 10% Fetal Bovine Serum (FBS, Sigma) and 2mM L-Glutamine (Sigma) at a density of 1E + 06/mL, and seeded 1 mL/tube in polypropylene tubes (Falcon). MoDCs were infected with STM-LT2 or STM-D23580 at a MOI of 10 and immediately spun down for 5 min to maximize bacteria-cell contact. This MOI was used, so that the frequency of infection was high enough to enable observation of infection events at the single-cell level in subsequent sorting experiments, while minimizing *Salmonella*-induced MoDCs death.

After incubation for 45 min at 37°C, extracellular bacteria were washed away with PBS. MoDCs were then incubated for 30 min in RPMI supplemented with 10% FBS, 2 mM L-glutamine and 100 µg/mL of gentamycin (MP Biomedicals) to kill extracellular bacteria. Gentamycin concentration was subsequently reduced to 30 µg/mL for the remainder of the experiment. At each time point, cells were lysed by addition of 500 µL of saponin 1% (w/v) (Sigma) in PBS followed by 5 min incubation at 37 °C. Cell lysates were serially diluted 10-fold in PBS and aliquots were plated onto LB agar. The number of intracellular bacteria was determined as CFU. The uptake of the bacteria was calculated as the number of bacteria recovered at 45 min p.i. divided by the initial number of bacteria. The percentage of bacterial survival was calculated as the number of bacteria recovered at each time point p.i. divided by the number of bacteria recovered at 75 min p.i. MoDCs viability was assessed by Tripan blue exclusion assay.

**Flow cytometry sorting**. At 2, 4 or 6 h post infection (p.i.), cells were harvested, washed, stained with Propidium Iodide (Biolegend) and re-suspended in FACS buffer containing PBS, 0.3% (v/v) bovine serum albumin (BSA, Sigma) and 2 mM EDTA (Invitrogen). Samples were immediately acquired on a FACSAria III sorter (BD Biosciences), applying fluorescence minus one controls to adjust compensation and sorting gates. Accuracy of a high purity single-cell sorting was confirmed using beads and cells. Live single cells were sorted into 96-well plates filled with 4 µL of lysis buffer (0.4% Triton, 2 U/µL RNase inhibitor (Clonotech, Takara), 7.5 µM oligo dT (Biomers) and 10 mM dNTP (ThermoFisher) [supplemented with 1:10,000,000 dilution of ERCC spike-in (ThermoFisher)]. Sorted cells were immediately vortexed, spun down and frozen on dry ice before storage at −80 °C. Bulk populations were sorted into Eppendorfs containing 10 µL of PBS, spun down, frozen on dry ice and stored at −80 °C.

**Quantification of intracellular bacteria**. To quantify the replication of intracellular bacteria, infected host cells were washed with PBS, re-suspended in FACS buffer, stained and FACS sorted as described before. At each time point, 10 single cells per each experimental condition were sorted in 8-well strips containing 1% saponin (Sigma) in PBS and plated onto agar plates to determine the number of intracellular CFU.

**Confocal microscopy**. MoDCs were plated in 8-well µ-slides (Ibidi) coated with 0.01% poly-L-Lysine (Sigma) at a density of 2E + 05 cells per well in 250 µL of RPMI supplemented with 10% FBS and 2 mM L-glutamine. Cells were infected with STM-LT2 or STM-D23580 and incubated for 2 h prior to fixation as described above.

Cells were washed twice with PBS, fixed with 4% paraformaldehyde (PFA; EMS) for 15 min and permeabilised with 0.05% Triton-X100 in PBS (Sigma) twice for 1min. Cells were blocked overnight at 4 °C with 250 µL blocking solution (5% human serum (Sigma), 5% FBS) per well.

The following day, MoDCs were incubated with anti-*Salmonella* CSA-1 FITC-conjugated antibody (KPL, Goat polyclonal, 1:100) in 140 µL of blocking solution for 1 h at RT. Following two washes with PBS for 1 min each, cells were incubated with 200 µL PBS containing 1 µg/mL DAPI (Thermo Fisher) for 10 min at RT. Finally, 10 µL of VECTASHIELD Antifade Mounting Medium (H-100, Vector laboratories) were added to each well.

Images were acquired on a Zeiss LSM 880 inverted confocal microscope with a Plan-Apochromat 63x/1.40 Oil objective and processed using the Zeiss Zen software and assembled in Adobe illustrator.

**Single-cell and bulk RNA-sequencing**. Single-cell RNA-seq libraries were prepared as per the Smart-seq2 protocol by Picelli et al.[17] with minor technical adaptations. Reverse transcription was carried on with 0.25 U/reaction of Super-Script II (Invitrogen) and PCR pre-amplification with 5′ biotinylated IS PCR primers (Biomers) for 22 cycles. PCR cleaning was performed by using Ampure XP Beads (Beckman Coulter) at a ratio 0.8:1 (beads:cDNA). cDNA was re-suspended in elution buffer (Qiagen) and assessed with a High-sensitivity DNA chip (Agilent).

For Bulk RNA-seq RNA was extracted from 5000 sorted cells by using the Qiagen RNeasy Micro Kit according to the manufacturer's instructions. One microliter of RNA was added to the respective wells in the plate containing lysis buffer and processed as for single cells with the exception of the PCR pre-amplification step that was carried out for 18 cycles. Barcoded Illumina sequencing libraries (Nextera XT, Library preparation kit, Illumina) were generated using the automated platform (Biomek FXp). Finally, libraries were pooled, and sequencing

was performed in paired-end mode for 2×75 bp cycles using the Illumina HiSeq 4000 sequencer.

**RNA-sequencing read mapping and expression quantification**. RNA-seq data were deposited in Gene Expression Omnibus under SuperSeries accession number GSE111546. Following deconvolution of RNA-seq raw reads into individual cells, initial quality assessment was performed using *FastQC* (v0.11.4). Reads pairs were mapped to a composite genome made by concatenating the human genome reference sequence (GRCh38) and 92 ERCC ExFold RNA Spike-In Mixes sequences (ThermoFisher). Alignment was performed using *HISAT2* (v2.0.3b) in default settings. Uniquely aligned read pairs were assigned and counted towards annotated human genomic genes and ERCC RNA spike-in features using *featureCounts* (v1.5.0-p2). All metrics were subsequently collected using *MultiQC* (v0.8) for reporting.

**Quality control of single-cell RNA-sequencing data**. Thirty-one cells (~8%) were identified as outliers and removed from all downstream analyses. Briefly, outliers were detected for each quality metric (Supplementary Figure 6) beyond an appropriate number of median absolute deviations (MADs) above, below or further than the median value of all single cells, according to the nature of the metric: GC content (±3 MADs), alignment rate (−3 MADs), library size (±3 MADs), percentage reads assigned to genes (−3 MADs), library complexity for the top 50, 100 and 200 genes, respectively (+3 MADs), count of detected genes (−2 MADs), ERCC content (+5 MADs), mitochondrial gene content (+3 MADs). Upon examination of endogenous genes, we observed similar distribution of average $log_{10}$-transformed read count per million (CPM) values across all fifteen experimental conditions (time point, infection, status). On average, 10,820 genes (range: 9698–12,143) were detected above an average 1 CPM, and 4221 genes (range: 3636–4827) were detected below an average 1 CPM, respectively (Supplementary Figure 7A).

In addition, we observed good correlation between the average $log_2$-transformed counts in cells from each condition and matched bulk samples prepared from 5000 cells sorted from MoDCs mock-infected or challenged with STM-LT2, STM-D23580 (Supplementary Figure 7B). Notably, although detected at very low abundance levels, ERCC spike-in features were more readily detected at higher read count levels in single cells relative to bulks, as expected from the higher relative proportion of ERCC spike-in molecules to endogenous RNA. We further confirmed a good correlation between the average counts of fragments yielded for each ERCC spike-in feature against the estimated count of molecules injected in each sample (Supplementary Figure 7C). ERCC spike-in molecules with an estimated count of injected molecules lower than one displayed an expected markedly larger variability and stochasticity of detection in individual cells. Overall, 8206 appreciably expressed endogenous genes were retained for downstream analyses (>10 CPM in ≥25% of cells of ≥1 experimental condition (Supplementary Figure 7D)).

**Normalization of single-cell RNA-sequencing data**. To account for the multifactorial nature of the experimental design and the considerable transcriptional changes expected over time, particularly in response to bacterial challenge, we estimated size factors for normalization from pools of cells sampled within unsupervised clusters. Specifically, four unsupervised clusters were identified by rank correlations between gene expression profiles using the *quickCluster* function of the *scran* package (v1.7.12). Notably, those unsupervised clusters largely recapitulated the three time points of the experimental design as well as the marked distinction between mock-infected and *Salmonella*-challenged MoDCs.

**Dimensionality reduction and pseudotemporal ordering**. Diffusion maps were computed using the *runDiffusionMap* wrapper function implemented in the *scater* package (v1.8.4), effectively computing a diffusion map from the 500 most variable genes in the data set using the *destiny* package (v2.10.2).

Pseudotemporal ordering was performed using the *monocle* package (v2.11.3). To isolate a set of ordering genes that define the progression of MoDCs through the infection challenge, we compared *Salmonella*-challenged cells collected at 2 and 6 h p.i. using the *differentialGeneTest* function. Next, we reduced data dimensionality using the *DDRTree* method and ordered cells along the resulting trajectory. Finally, we identified genes that change as a function of the computed pseudotime using the *differentialGeneTest* function. A *q*-value threshold of 0.001 was applied to all differential expression tests in the *monocle* workflow. Gene expression heat maps were produced using the *ComplexHeatmap* package (v1.18.1).

**Identification of single-cell clusters**. To identify clusters of cells in transcriptionally distinct states while controlling for the pronounced dynamics of gene expression over time, unsupervised clusters were computed separately at each time point by rank correlation between gene expression profiles using the *quickCluster* function of the *scran* package.

**Single-cell differential expression analysis**. The *scde* package (v2.7.0) was used to estimate differential expression between groups of cells. Error models were

computed separately for cells at each time point, while determining cross-fits and expected expression magnitudes separately within each experimental condition. The prior distribution for gene expression magnitudes was estimated separately for each time point using the *scde.expression.prior* function in default settings. *Z* scores returned by *scde* were converted to empirical *P*-values using the *pnorm* function of the R *stats* package (v3.5.0), and a significance cut-off $P < 0.01$ was applied to define differentially expressed genes.

**Quality control of small bulk RNA-sequencing data**. Outliers were detected for each quality metric as described above for single-cell RNA-sequencing libraries. Multiple RNA-seq libraries from donor 3 appeared as outliers in multiple quality control metrics (Supplementary Figure 16). Consequently, all libraries generated from this donor were excluded to ensure a balanced experimental design for downstream differential expression analyses.

Overall, 8867 genes expressed on average above 10 CPM in at least one experimental condition (across the remaining two donors) were retained for downstream analysis.

**Small bulk differential expression analysis**. The *DESeq2* package (v1.20.0) was used to estimate differential expression between experimental groups using the Negative Binomial GLM fitting and Wald statistics[70]. Size factors were estimated using the median ratio method, and dispersion estimates for the Negative Binomial distributed data were computed using a parametric dispersion-mean fit.

**Quantitative PCR**. MoDCs were stimulated with either STM-LT2 or STM-D23580 or left unstimulated. At 4 and 6 h p.i. single cells were FACS sorted in lysis buffer and cDNA was prepared according to Smart-seq2 protocol as described above. cDNA from eight individual cells from each experimental group was pooled and qPCR for selected genes was performed. qPCR reaction was carried on in 96-well-plate in 20μL final volume containing: Dual Lock DNA pol Master Mix (Life Technologies), forward and reverse primers (Supplementary Table 4) at a final concentration of 1 μM each, H$_2$O and 10 ng of cDNA. After the initial denaturation steps at 50 °C for 2 min and 95 °C for 2 min, PCR was performed for 40 cycles (95 °C for 3 s and 60 °C for 30 s for each cycle) by using the Quanti7 Studio machine. The specificity of primers was confirmed using Primer- BLAST (NCBI). Fold changes were determined using the $2^{-\Delta Ct}$ method. The mRNA levels were expressed in relative copy numbers normalized against the Glyceraldehyde 3-phosphate dehydrogenase (GAPDH) mRNA.

**ELISAs**. IL-1β, IL-12p40 and IL-10 levels were measured from culture supernatants by ELISA kits (R&D System) according to manufacturer's instructions. The range for IL-1β was 250–3.91 pg/mL, for IL-12p40 was 4000–61.5 pg/mL, and for IL-10 was 2000–31.3 pg/mL. The results were expressed as pg/mL for each cytokine.

**Fluorescence-activated cell sorting (FACS) analysis**. MoDCs were stimulated with either STM-LT2 or STM-D23580 or left unstimulated. At specific time points post infection, MoDCs were washed and incubated for 30 min with the following antibodies: anti-CD83 (PE; HB15e), anti-CD86 (PeCy7; IT2.2), anti-HLA-DR (APC; L243, or PerCP ef710; L243), HLA-A,B,C (APC Fire™ 750, w6/32) and anti-CD1a (FITC, HI149), all obtained from BioLegend. Zombie Aqua™ fixable viability dye (BioLegend) was used for exclusion of dead cells. After incubation, cells were washed in FACS buffer and fixed in 2% PFA. Samples were acquired on a Fortessa X20 flow cytometer (BD Biosciences) and files were analysed on Flowjo (v.10.4.1).

**T cell clone generation**. *Salmonella*-specific CD4$^+$ T cell clones were generated and expanded as previously described[47]. Briefly, CCR7$^-$CD38$^+$CD4$^+$ T cells, isolated from volunteers after diagnosis of enteric fever, were sorted as single cells, expanded in vitro with phytohemagglutinin (PHA; Remel) and IL-2 in the presence of irradiated feeders for at least 3 weeks. Expanded clones were tested for their capacity of recognizing autologous Epstein Barr Virus (EBV)-transformed lymphoblastoid cell lines (EBV-LCL), either infected with live *Salmonella* or pulsed with *Salmonella*-related peptide pools (PhoN or HlyE). T cell clones were considered positive with IFN-γ > 5% in the presence of bacteria infected autologous cells, and/or IFN-γ > 20% in the presence of cells pulsed with proteins or peptide pools.

*HLA-DRB1*-restricted PhoN-specific CD4$^+$ T cell clones cross-reactive against *S.* Typhimurium and *S.* Typhi serovars, as well as HlyE-specific CD4$^+$ T cell clones reactive against *S.* Typhi, were used in this study.

**Antigen presentation assay and intracellular staining**. Human MoDCs were generated as described above, from CD14$^+$ peripheral monocytes obtained from healthy blood donors, and screened for matching *HLA-DRB1* haplotypes with the selected T cell clones. At 6 h p.i., MoDCs were stained for 30 min with anti-CD83 (PE; HB15e) and Zombie Aqua™ fixable viability dye to exclude dead cells. Cells were washed, re-suspended in FACS buffer and immediately sorted on a FACSAria III instrument. Infected cells were sorted based on their CD83 surface expression: namely, STM-D23580-infected MoDCs were sorted as CD83$^{-/low}$ or CD83$^{high}$,

while STM-LT2 and ST-Ty2 infected MoDCs were sorted as CD83$^+$. Uninfected MoDCs were used as negative control.

Sorted live infected MoDCs were washed and co-cultured with *Salmonella*-specific CD4$^+$ T cell clones, at a ratio of 1:6 (MoDCs: T cells) in U-bottom 96-well plates (Costar), in RPMI supplemented with 10% FBS, 2 mM L-Glut, 10 mM HEPES (Life Technologies), 1 mM Sodium Pyruvate (Sigma), 0.01 mM MEM Non Essential Amino acids (Life Technologies) and 50 μg/mL of gentamycin (MP Biomedicals). To enhance intracellular cytokine signals, 5 ng/μL of Brefeldin A (BioLegend) were added at 2 h after the beginning of co-culture. Following 14 h incubation, T cells were fixed, permeabilized and stained with the following antibodies: anti-CD3 (Alexa 700, UCHT1; BioLegend), anti-CD4 (APC ef780, RPA-T4; eBioscience), anti-CD40L (FITC, 24–31; BioLegend); anti TNF-α (PECy7, Mab11; BioLegend), anti- IFN-γ (PE Dazzle, 4 S.B3; BioLegend). Viability was assessed with live/dead Aqua staining. Cells were acquired on a Fortessa X20 flow cytometer and data were analysed on FlowJo (v.10.4.1).

**Statistical analysis**. Statistical analyses were performed using GraphPad-Prism7 (GraphPad Software—San Diego, CA, United States). Pairwise comparisons were calculated using paired two-tailed Student's *t*-test. Differences among groups were determined by one-way or two-way ANOVA, as appropriate. ANOVA test statistics were corrected post-hoc by Tukey, applying a 95% confidence interval. A *P*-value < 0.05 was considered statistically significant. All other analyses were performed in the *R* statistical programming environment, using the latest version of *R* and *Bioconductor* packages as appropriate.

## Data availability
Sequences data have been deposited in the Gene Expression Omnibus under SuperSeries code GSE111546. The computational workflow has been made available at https://doi.org/10.5281/zenodo.1217158. All experimental data are available from the authors. A reporting summary for this Article is available as a Supplementary Information file.

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

## Acknowledgements

A.A. and G.N., were funded by UK Medical Research Council (MRC). L.P.-L. was supported by the MRC and Oxford NIHR biomedical Research Centre. A.S. was supported by the MRC, an NIHR Research Professorship and Wellcome Trust Investigator

Award. We would like to acknowledge Paul Sopp and Craig Waugh in the flow cyto-metry facility at the Weatherall Institute of Molecular Medicine. The facility is supported by the MRC HIU; MRC MHU (MC_UU_12009); NIHR Oxford BRC and John Fell Fund (131/030 and 101/517), the EPA fund (CF182 and CF170) and by the WIMM Strategic Alliance awards G0902418 and MC_UU_12025. We acknowledge support of the Oxford Single Cell Consortium and thank the Oxford Genomics Centre at the Wellcome Centre for Human Genetics (funded by Wellcome Trust grant reference 203141/Z/16/Z) for the generation and initial processing of the sequencing data. We thank Life Science Editors for editing assistance.

## Author contributions

A.A. and L.P.-L. conceived, designed and performed experiments. K.R.A. performed computational analyses. A.C. and D.S. provided support for computational analyses. L. P.-L. performed flow cytometry experiments. N.A. assisted in single-cell sequencing experiments. G.N. assisted with experimental design and manuscript editing. J.K. and C. L. performed confocal microscopy experiments. T.A. provided reagents. A.A., K.R.A. and A.S. wrote the manuscript. M.G. provided advice. A.S. conceived the study, obtained funding and supervised the work.

## Additional information

**Competing interests:** The authors declare no competing interests.

