## [Peer Review File · Nature Communications]

Reviewers' comments:

Reviewer #1 (Remarks to the Author):

Major Points

The authors have here used monocyte-derived cells to investigate the biological effects of infection with invasive or non-invasive salmonella. This is important work, and the data generated are both novel and interesting. The manuscript, however, consistently refers to DCs - which are not monocyte-derived cells. It should be re-focused so that these interesting data are described and discussed in the context of the monocyte-derived cells that were actually used for the experiments.

When describing the potentially-important differences between cells exposed to STM-LT2 and STM-D23580 in Figure 3A and B, it would be helpful if the statistical significance of any observed differences could be described. To my eye, the cytokine expression patterns in 3A appear to indicate that all the groups are heterogeneous, and that a proportion of STM-LT2 infected or exposed cells have very similar gene expression to the STM-D23580 cells. The real story might therefore be that one strain was better at invading than the other, for artifactual technical reasons. In the few hundred cells chosen for sequencing, rather than there being intrinsic differences in the properties of the two salmonella strains.

As many differences between cells are not apparent at 4h, and as the exposed and infected cells are cultured together in the same wells for the entire period of the experiment, there are many biological reasons that could contribute to the differences seen other than differences between the PAMPS expressed by the different bacterial strains. For instance a very small difference in the MOI between wells, or a small difference in the proportions of cells dying in the cultures, amplified over the period of the experiment, could lead to the kind of differences observed here.

The data in Figure 4 are insufficient to conclude that STM-D23580 induces a 'tolerogenic' profile. The differences involve only a small number of genes, and their combined functional effect has not been assessed. The results from Figures 5, 6, and 7 do not address this issue.

The data in Figures 5, 6, and 7 appear to show that, in contrast to STM-LT2-infected cells, a population of STM-D23580-infected cells express less CD83, and that these CD83-low cells are less able to induce TNFalpha and IFN-gamma production (and Cd40L expression) from a T cell clone. However, the CD83+ STM-D23580-infected cells, which compose the majority of the STM-D23580-infected cells, show no such defect. These data are potentially interesting, but as the CD83 reduction that is associated with the loss of function only occurs in a minority of the STM-D23580-infected cells, it's biological importance appears incompatible with the authors' interpretation that, for instance, the "...invasive Salmonella ... dramatically modulates the ability of DCs to trigger antigen-specific T cell activation."

Minor Points

The data in supplementary figure 2 indicate that only 2% or fewer STM-exposed DCs acquire detectable fluorescence. The proportion of infected cells is lower than would be expected for an in vitro infection of monocyte-derived cells. It would be helpful if the authors could show a simple immunofluorescent image of their infected culture, demonstrating that a low frequency of cells are infected. The alternative explanation is that a high proportion of cells may be infected but that the level of fluorescence is insufficient for them to be separated from the 'exposed' cells by flow cytometry. If this latter is the case, it will have a significant effect on the interpretation of the subsequent experiments.

A rough calculation using the data from Figure S3 suggests that at the 4h time point the majority of bacteria are contained within the exposed rather than the infected population (0.5 cfu x 92% exposed >> 5 cfu x 1.5% infected). This will have some effect on the author's ability to distinguish

between the infected and exposed populations, and should be acknowledged in the manuscript.

From Figure 2A, the authors' claim (l133) that there are 'notable' differences between infected and exposed cells at later time points. The way this figure is presented should be improved so these differences are easier for the reader to discern, and the specific differences of interest should them be precisely described. I, for one, am unable to see any notable differences between the exposed and infected clusters at 6h, for instance.

In describing Figure 2B, a clearer explanation is required to explain of how the data support the conclusion that "the invasive STM-D23580 induces more divergent gene expression differences between infected and exposed DCs."

When describing data, for example in lines 149-154 the authors should make it unambiguously clear when they are referring to gene expression data, and when they are describing protein production.

Data in S7 show differences between LPS-stimulated and HK-STM-stimulated cells at 6h, but not differences between the different types of HK-STM. At the later time points, the majority of concentrations do not show any difference between HK-STM types. These data do not really support the idea that STM-D23580 generates a "hyperactivated" immune state.

Reviewer #2 (Remarks to the Author):

In the manuscript presented by Aulicino et al, the authors have infected human dendritic cells (DCs) with a virulent Salmonella strain ST313 and a less invasive one (STM-LT2) and studied the transcriptomics response of the host using single-cell RNA-seq. The strain ST313 is highly relevant in clinical settings since it emerges as a multi-drug resistant strain. The authors looked at the host response between 0/2/4 and 6 hr post-infection. Overall the authors want to understand how the response of DCs differ after the infection with the two strains.

The experimental design and data collection are robust and comprehensively described. The use of single-cell RNA-seq is remarkable and very well performed. This study is a first and the comparison between the two Salmonella strains brings a unique dimension to the work. The overall scope and ambition of the study is of prime interest.

This said I find the way that the authors displayed the results and subsequently the message of the publication difficult to follow. The importance of the publication is shadowed by many details and the overall quality of the paper is unfortunately decreased by figures of poor quality. Many genes names are quoted across the manuscript but how these are united in not clearly stated. Since the study is highly multidimensional (15 conditions and single-cell), and the differences across the conditions seems narrow, I understand the challenge faced by the authors. At the end, how the response of DCs change between the two strains is not clear for me. How the transcriptome change between infected and bystander cells remain obscure.

A good example that the authors could follow is Shalek et al 2014 (Nature) that has unified differential gene expression into co-regulated signatures- that could help resolving the complexity of the data and avoid list of genes. Also, instead of working with the merged data, one can consider studying the different conditions in a separated manner and only merge when needed. The authors should try to implement temporal single-cell RNA-seq analysis like monocle or diffusion maps. Also, the authors should not overstate the conclusions from the data or provide clear support for them (see below specific details).

- Figure 1 and Figure 2 poorly display the changes happening between the different conditions and should be deeply revised. In detail:

- the scheme Figure 1A indicate 2 bacteria per "infected" cells: is that true? The "Exposed" cells harbor a cell on the membrane: it is confusing because "exposed" cells should have no bacteria neither intracellular nor extracellular (as shown S-fig 2)
 - Figure 1C: the heatmap in the current version cannot be read and is not informative on the changes occurring between the different states. The scale of the color is not even indicated and the heatmap is not clustered in y axis.
 - Since the study encompass many different conditions (2 bacterial strains, 3 conditions, 4 time-points), figure 1 and figure 2 should be revised to better highlight the transcriptomics changes happening. For example, the authors could have a separate t-SNE for each bacteria strain.
 - Figure 2A state the presence of clusters: are these clusters identified in a unsupervised manner (as written lines 128-132 and line 157-8)? The histogram fig. 2A is cryptic : how to understand such a graph?
 - Figure 2B is extremely difficult to read. Instead, the authors should use a heatmap and this analysis seems in clear conflict with Figure 4 that address the same question (strain comparison).
 - The SCDE analysis comes by far too late on Figure 4 and what is exactly the difference with Figure 2B?
 - Figure 4: what are x and y axis?
- The difference between the two strains is restricted to low expressed genes. Are those changes relevant statistically? The authors should use a heatmap so we can better look the differences between the invasive and non-invasive strain.
- The authors demonstrate that DC-mediated antigen presentation to CD4+ T cells is impaired (lines 247 to 270). And consequently claim at the end that: "Our results further delineate a molecular pathway exploited by bacteria to evade immediate detection and enable dissemination in immune compromised hosts." In immune compromised hosts CD4 are dramatically reduced. So, how can such claim be made in such a strong manner?
 - Line 377-8: how can such study help designing vaccines? Is such statement supported by data?
 - Suppl Figure 11: the PCA plot does not display the hours of infection. Is it a single-time-point infection?

Minor comments:

- In the introduction, the authors should provide the properties and some background information on STM-LT2
- I encourage the authors to perform an image of the infected cells using fluorescence microscopy.
- Between 0 and 6hr, does the bacteria replicate in DCs? The authors should add a sentence to clarify the replication status of the bacteria.
- The word "exposed" should be clarified because the text hesitate between "Exposed" and "Bystander" (like in the abstract). Maybe "Bystander" is a better choice.
- Page 4 line 76: is reference 22 is not related to "Salmonella growth" and should be removed from here or better explained.
- Suppl Figure 4: the accumulation of panels makes the figure difficult to read; one can thing maybe to have a summary figure that wraps up all the content,
- Suppl Figure 4-5: what is the meaning of "feature"?

Referee

If the authors needs to contact me to discuss the data or specific points of my reviews, I encourage them to do so. I have no conflict of interest with this work.

Antoine-Emmanuel Saliba, PhD
 Group leader, Single-Cell Analysis
 Helmholtz Institute for RNA-based infection research (HIRI)
 University of Würzburg

Josef-Schneider-Str. 2/D15
 D-97080 Würzburg

Germany

Phone: 49(0)931-3181341

Email: emmanuel.saliba@helmholtz-hiri.de

Response to reviewers for Aulicino *et al.* “Invasive Nontyphoidal Salmonella Exploits Divergent Immune Evasion Strategies in Infected and Bystander Dendritic Cells”

We would like to thank both reviewers for their insightful comments which have very much improved our revised manuscript.

Reviewer #1 (Remarks to the Author):

Major Points:

1) *The authors have here used monocyte-derived cells to investigate the biological effects of infection with invasive or non-invasive salmonella. This is important work, and the data generated are both novel and interesting. The manuscript, however, consistently refers to DCs - which are not monocyte-derived cells. It should be re-focussed so that these interesting data are described and discussed in the context of the monocyte-derived cells that were actually used for the experiments.*

We agree with the reviewer that the cell type used in our work should be more clearly stipulated and have substituted “Monocyte derived dendritic cells (MoDCs) for DCs, throughout the text.

2) *When describing the potentially-important differences between cells exposed to STM-LT2 and STM-D23580 in Figure 3A and B, it would be helpful if the statistical significance of any observed differences could be described. To my eye, the cytokine expression patterns in 3A appear to indicate that all the groups are heterogeneous, and that a proportion of STM-LT2 infected or exposed cells have very similar gene expression to the STM-D23580 cells. The real story might therefore be that one strain was better at invading than the other, for artefactual technical reasons. In the few hundred cells chosen for sequencing, rather than there being intrinsic differences in the properties of the two salmonella strains. As many differences between cells are not apparent at 4h, and as the exposed and infected cells are cultured together in the same wells for the entire period of the experiment, there are many biological reasons that could contribute to the differences seen other than differences between the PAMPS expressed by the different bacterial strains. For instance a very small difference in the MOI between wells, or a small difference in the proportions of cells dying in the cultures, amplified over the period of the experiment, could lead to the kind of differences observed here.*

Figure 3A (in the first submission; now Figure 4 in the revised manuscript) has now been amended to display the significance of differences between the conditions in Figure 4. In addition we have provided an extra supplementary data table that lists differentially expressed genes between the different conditions.

We found STM-LT2 and STM-D23580 demonstrated equivalent ability to survive and multiply within MoDCs. No significant differences were observed in the number of intracellular CFU detected with either bacterial strain at any of the time points examined (supplementary figure 4A). The number of bacteria of either strain was assessed at each time point post-infection by plating 10-fold dilutions of bacterial suspension, in quadruplicate, on LB agar. The number of bacteria was determined as Colony Forming Units (CFU) according to the Miles and Misra method as shown in Fig S4A.

The percentage uptake and survival within MoDCs was also comparable for both strains and demonstrated equivalent intracellular growth over time (supplementary figure 4B). The percentage uptake of bacteria was calculated as the number of bacteria recovered at 45 min post-infection divided by the initial number of bacteria (supplementary figure 4C). The percentage survival of either strain was calculated as the number of bacteria recovered at each time point post-infection divided by the number of bacteria recovered at 75min post-infection. Of note, no significant differences were observed in the viability of MoDCs infected with the two bacterial strains during the course of either infection (supplementary figure 4D). In order to limit MOI discrepancies and to ensure consistency, all the experiments presented in this work were performed using frozen live bacterial stocks. Over the period of time in which these experiments were carried out, the number of viable bacteria (CFU/ml) in each batch of stock was carefully

and frequently monitored (see Methods section). We are therefore confident each bacterial strain invaded MoDCs equivalently.

It is correct that there could be other reasons other than differential PAMP expression between the strains that might account for the differences in gene expression observed, however differential PAMP expression likely contributes. We have amended the text to convey this point accordingly.

3) *The data in Figure 4 are insufficient to conclude that STM-D23580 induces a 'tolerogenic' profile. The differences involve only a small number of genes, and their combined functional effect has not been assessed. The results from Figures 5, 6, and 7 do not address this issue.*

We agree with this reviewer the results of Figure 4 (in the first submission; now Figure 5 in the revised manuscript) could be better described to more accurately reflect our findings. We have amended the text removing description of STM-D23580 inducing a 'tolerogenic' profile instead describing the molecular immune evasion strategy of the invasive strain more accurately.

4) *The data in Figures 5, 6, and 7 appear to show that, in contrast to STM-LT2-infected cells, a population of STM-D23580-infected cells express less CD83, and that these CD83-low cells are less able to induce TNFalpha and IFN-gamma production (and Cd40L expression) from a T cell clone. However, the CD83+ STM-D23580-infected cells, which compose the majority of the STM-D23580-infected cells, show no such defect. These data are potentially interesting, but as the CD83 reduction that is associated with the loss of function only occurs in a minority of the STM-D23580-infected cells, it's biological importance appears incompatible with the authors' interpretation that, for instance, the "...invasive Salmonella ... dramatically modulates the ability of DCs to trigger antigen-specific T cell activation."*

While it is correct only a proportion of MoDCs showed low CD83 expression, this proportion accounted for up to 35% of STM-D23580 challenged MoDCs. By way of example, CD83 histograms from 3 independent donors infected with the two strains are shown below. Recently published human challenge models of *Salmonella* (Dobinson *et al.*, 2017), have confirmed that an oral dose of 10^3 CFU from invasive *Salmonella* strains is sufficient to cause disease in healthy adults. Taking into account that STM-D23580 infections are highly prevalent among immunocompromised individuals, a 30% effect on the *MARCH1/CD83/IL10* axis is likely to contribute to inefficient bacterial clearance in these individuals. There are many examples of immune evasion strategies employed by pathogens in antigen presenting cells where incomplete effects are observed for example with HIV-1 Nef, that induces a small percentage downregulation in surface MHC I and CD4 to evade adaptive immunity (Dikeakos *et al.*, 2010 PMID:20702582).

Donor 1

Donor 2

Minor points:

5) The data in supplementary figure 2 indicate that only 2% or fewer STM-exposed DCs acquire detectable fluorescence. The proportion of infected cells is lower than would be expected for an *in vitro* infection of monocyte-derived cells. It would be helpful if the authors could show a simple immunofluorescent image of their infected culture, demonstrating that a low frequency of cells are infected. The alternative explanation is that a high proportion of cells may be infected but that the level of fluorescence is insufficient for them to be separated from the 'exposed' cells by flow cytometry. If this latter is the case, it will have a significant effect on the interpretation of the subsequent experiments.

Supplementary figure 3 validates our flow cytometry gating strategy and demonstrates how viable bacteria were almost uniquely recovered from the "infected" gate. Previous literature also stipulates only a low proportion of Dendritic cells internalize bacteria:

- Yrlid *et al.* Infect Immun. 2001 (1%-10% of CD11c+ cells are infected with *S. typhimurium*)
- Hopkins *et al.* Cell Microbiol. 2000 (<1% of splenic DCs are infected with *S. typhimurium*)

- Jiao *et al.*, J Immunol 2002 (~2% of splenic DCs contain *M. bovis* BCG)
- Cruz-Adalia *et al.*, Cell 2014 (~2% of MoDCs contain *L. monocytogenes*, ~2% of MoDCs contain *S. enteritidis*, ~5% of MoDCs contain *S. aureus*)

As suggested by the reviewer, we have provided a confocal microscopy image demonstrating the frequency of infected MoDCs for our work (Figure Supplementary 3 in the revised manuscript).

In contrast to the flow cytometric setup, the configuration of our microscope did not allow us to clearly detect the APC (or BV412) fluorescence signal of the CellTrace dye used in the study. As visible from Fig S3 panel A, the signal from the bacterial staining, whereas visible, is not optimal. Therefore, we performed confocal using a specific primary anti-*Salmonella* antibody (CSA-1, BacTrace). The staining with the *Salmonella* antibody requires cell permeabilization and thus is not compatible with Single cell RNA-sequencing experiments. Low frequency of infected cells can be observed for each donor (A-B Donor 1; C Donor 2).

6) A rough calculation using the data from Figure S3 suggests that at the 4h time point the majority of bacteria are contained within the exposed rather than the infected population ($0.5 \text{ cfu} \times 92\% \text{ exposed} \gg 5 \text{ cfu} \times 1.5\% \text{ infected}$). This will have some effect on the author's ability to distinguish between the infected and exposed populations, and should be acknowledged in the manuscript.

In this revised manuscript we have replaced the term “exposed” with “bystander”, as suggested by reviewer 2. Supplementary figure 3 (in the first submission; now Supplementary Figure 4 in the revised manuscript) was generated by sorting 10 Bystander MoDCs and 10 Infected MoDCs (3 donors, 8 replicates each), lysing them and plating for CFU. Therefore the CFU measured are derived from the same number of Infected and Bystander MoDCs.

When the same number of Infected and Bystander cells were sampled, the CFU ratio (CFU/MoDC) was kept constant: ~0,05 in Bystander and 0.5 in Infected. As reported in Supplementary table 3, an equal number of cells per condition (Uninfected, Infected, Bystander) were used for comparison. Bulk cell experiments, in which nearly 1 million MoDCs were lysed and plated, confirmed our findings. As shown in the table below, the Real MOI calculated by dividing the number of CFU by the number of live MoDCs in the well, was comparable to the Real MOI obtained from 10 Infected sorted cells.

Bulk experiment:

Intracellular CFU						
	STM-LT2			STM-D23580		
	Donor 1	Donor 2	Donor 3	Donor 1	Donor 2	Donor 3
2h p.i.	2.10E+05	1.20E+05	1.93E+05	2.50E+05	2.60E+05	2.93E+05
4h p.i.	3.50E+05	1.80E+05	2.55E+05	2.80E+05	2.90E+05	3.03E+05
6h p.i.	2.80E+05	2.50E+05	3.28E+05	3.20E+05	3.00E+05	4.40E+05
Live MoDCs						
	Challenged with STM-LT2			Challenged with STM-D23580		
	Donor 1	Donor 2	Donor 3	Donor 1	Donor 2	Donor 3
2h p.i.	1.00E+06	1.00E+06	6.48E+05	8.00E+05	8.00E+05	5.80E+05
4h p.i.	7.50E+05	8.70E+05	4.50E+05	8.00E+05	9.30E+05	5.00E+05
6h p.i.	6.80E+05	6.70E+05	7.00E+05	9.20E+05	9.00E+05	8.00E+05
Real MOI						
	Num. STM-LT2 CFU/Live MoDCs			Num. STM-D23580 CFU/Live MoDCs		
	Donor 1	Donor 2	Donor 3	Donor 1	Donor 2	Donor 3
2h p.i.	0.21	0.12	0.30	0.31	0.33	0.51
4h p.i.	0.47	0.21	0.57	0.35	0.31	0.61
6h p.i.	0.41	0.37	0.47	0.35	0.33	0.55

Sorted MoDCs

Intracellular CFU in 10 sorted STM-LT2 infected MoDCs (gated on live)																
	Donor 1								Donor 2							
	Rep.1	Rep.2	Rep.3	Rep.4	Rep.5	Rep.6	Rep.7	Rep.8	Rep.1	Rep.2	Rep.3	Rep.4	Rep.5	Rep.6	Rep.7	Rep.8
2h	3	4	2	2	3	3	1	2	2	2	5	2	3	3	2	2
4h	4	4	0	3	6	3	2	3	2	3	2	4	4	3	2	3
6h	2	6	5	6	4	5	2	4	6	3	3	4	4	0	4	3
Real MOI (Num. STM-LT2 CFU/10)																
	Donor 1								Donor 2							
	Rep.1	Rep.2	Rep.3	Rep.4	Rep.5	Rep.6	Rep.7	Rep.8	Rep.1	Rep.2	Rep.3	Rep.4	Rep.5	Rep.6	Rep.7	Rep.8
2h	0.3	0.4	0.2	0.2	0.3	0.3	0.1	0.2	0.2	0.2	0.5	0.2	0.3	0.3	0.2	0.2
4h	0.4	0.4	0	0.3	0.6	0.3	0.2	0.3	0.2	0.3	0.2	0.4	0.4	0.3	0.2	0.3
6h	0.2	0.6	0.5	0.6	0.4	0.5	0.2	0.4	0.6	0.3	0.3	0.4	0.4	0	0.4	0.3

Intracellular CFU in 10 sorted STM-LT2 Bystander MoDCs (gated on live)																
	Donor 1								Donor 2							
	Rep.1	Rep.2	Rep.3	Rep.4	Rep.5	Rep.6	Rep.7	Rep.8	Rep.1	Rep.2	Rep.3	Rep.4	Rep.5	Rep.6	Rep.7	Rep.8
2h	0	0	0	0	0	0	2	1	0	0	1	0	0	0	0	1
4h	0	0	0	0	0	3	0	0	0	0	0	0	0	0	2	0
6h	0	2	0	0	0	0	1	0	1	0	0	0	0	0	0	0
Real MOI (Num. STM-LT2 CFU/10)																
	Donor 1								Donor 2							
	Rep.1	Rep.2	Rep.3	Rep.4	Rep.5	Rep.6	Rep.7	Rep.8	Rep.1	Rep.2	Rep.3	Rep.4	Rep.5	Rep.6	Rep.7	Rep.8
2h	0	0	0	0	0	0	0.2	0.1	0	0	0.1	0	0	0	0	0.1
4h	0	0	0	0	0	0.3	0	0	0	0	0	0	0	0	0.2	0
6h	0	0.2	0	0	0	0	0.1	0	0.1	0	0	0	0	0	0	0

Intracellular CFU in 10 sorted STM-D23580 infected MoDCs (gated on live)																
	Donor 1								Donor 2							
	Rep.1	Rep.2	Rep.3	Rep.4	Rep.5	Rep.6	Rep.7	Rep.8	Rep.1	Rep.2	Rep.3	Rep.4	Rep.5	Rep.6	Rep.7	Rep.8
2h	5	5	4	7	6	4	0	6	7	1	5	4	4	3	2	6
4h	4	1	5	2	3	4	5	4	7	5	7	5	7	6	7	4
6h	5	6	7	3	7	6	5	4	9	3	5	6	4	6	8	7
Real MOI (Num. STM-D23580 CFU/10)																
	Donor 1								Donor 2							
	Rep.1	Rep.2	Rep.3	Rep.4	Rep.5	Rep.6	Rep.7	Rep.8	Rep.1	Rep.2	Rep.3	Rep.4	Rep.5	Rep.6	Rep.7	Rep.8
2h	0.5	0.5	0.4	0.7	0.6	0.4	0	0.6	0.7	0.1	0.5	0.4	0.4	0.3	0.2	0.6
4h	0.4	0.1	0.5	0.2	0.3	0.4	0.5	0.4	0.7	0.5	0.7	0.5	0.7	0.6	0.7	0.4
6h	0.5	0.6	0.7	0.3	0.7	0.6	0.5	0.4	0.9	0.3	0.5	0.6	0.4	0.6	0.8	0.7

Intracellular CFU in 10 sorted STM-D23580 Bystander MoDCs (gated on live)																
	Donor 1								Donor 2							
	Rep.1	Rep.2	Rep.3	Rep.4	Rep.5	Rep.6	Rep.7	Rep.8	Rep.1	Rep.2	Rep.3	Rep.4	Rep.5	Rep.6	Rep.7	Rep.8
2h	0	0	0	0	0	0	0	1	0	0	0	0	0	0	0	0
4h	0	0	0	0	0	1	1	0	0	1	0	0	0	0	0	0
6h	0	0	1	0	0	0	0	3	0	0	2	0	0	0	0	0
Real MOI (Num. STM-D23580 CFU/10)																
	Donor 1								Donor 2							
	Rep.1	Rep.2	Rep.3	Rep.4	Rep.5	Rep.6	Rep.7	Rep.8	Rep.1	Rep.2	Rep.3	Rep.4	Rep.5	Rep.6	Rep.7	Rep.8
2h	0	0	0	0	0	0	0	0.1	0	0	0	0	0	0	0	0
4h	0	0	0	0	0	0.1	0.1	0	0	0.1	0	0	0	0	0	0
6h	0	0	0.1	0	0	0	0	0.3	0	0	0.2	0	0	0	0	0

On average, the Real MOI measured in a bulk experiment correspond to the Real MOI measured with Infected sorted cells:

Real MOI					
Bulk		Sorted live Infected MoDCs		Sorted live Bystander MoDCs	
STM-LT2	STM-D23580	STM-LT2	STM-D23580	STM-LT2	STM-D23580
0.21	0.38	0.25	0.43	0.03	0.006
0.41	0.42	0.30	0.47	0.03	0.018
0.42	0.41	0.38	0.56	0.02	0.03

7) From Figure 2A, the authors' claim (1133) that there are 'notable' differences between infected and exposed cells at later time points. The way this figure is presented should be improved so these differences are easier for the reader to discern, and the specific differences of interest should them be precisely described. I, for one, am unable to see any notable differences between the exposed and infected clusters at 6h, for instance.

In describing Figure 2B, a clearer explanation is required to explain of how the data support the conclusion that "the invasive STM-D23580 induces more divergent gene expression differences between infected and exposed DCs."

In this revised manuscript we have updated the figures to better relay the key findings of our data. We replaced Figure 2A-C with heat maps and diffusion maps that are inherently more suitable for illustrating the differentiation trajectory of cells challenged with bacteria, reflecting the sequential time points of the experiment, as well as the unsupervised clusters at each time point.

Also, we have included supplementary tables documenting the sets of analyses performed between conditions. These describe differential expression and gene ontology analyses performed for 1) direct comparisons between all 5 treatment groups at each time point and 2) comparisons between individual clusters and all other clusters for the identification of 'marker' genes.

8) When describing data, for example in lines 149-154 the authors should make it unambiguously clear when they are referring to gene expression data, and when they are describing protein production.

We have followed the HUGO Gene nomenclature Committee guidelines, responsible for providing human gene names in our work. In the manuscript human gene symbols are italicised, with all letters in uppercase, while protein designations are the same as the gene symbol, but are not italicised, with all letters in uppercase. In addition we have stipulated gene or protein expression where appropriate.

9) Data in S7 show differences between LPS-stimulated and HK-STM-stimulated cells at 6h, but not differences between the different types of HK-STM. At the later time points, the majority of concentrations do not show any difference between HK-STM types. These data do not really support the idea that STM-D23580 generates a "hyperactivated" immune state.

Supplementary figure 7 has now been moved to supplementary figure 9A and 9B. We agree while the changes observed are significant, the use of the word "hyperactivated" is poorly representative. We have changed the description of the phenotype observed from "hyperactivated" to "enhanced"

The main text has been amended in the following way:

"The enhanced pro-inflammatory phenotype of MoDCs exposed to STM-D23580 compared to STM-LT2 raises the possibility that STM-D23580 surface PAMPs might exert a more potent pro-inflammatory response compared to STM-LT2. To test this hypothesis, we exposed MoDCs to heat-killed (HK) Salmonella treated with DNase, RNase and Proteinase K, and assayed pro-inflammatory cytokine responses. We found higher IL-1 β and IL-12p40 secretion by MoDCs stimulated with HK STM-D23580 as compared to HK STM-LT2"

Reviewer #2 (Remarks to the Author):

Major points:

In the manuscript presented by Aulicino et al, the authors have infected human dendritic cells (DCs) with a virulent Salmonella strain ST313 and a less invasive one (STM-LT2) and studied the transcriptomics response of the host using single-cell RNA-seq. The strain ST313 is highly relevant in clinical settings since it emerges as a multi-drug resistant strain. The authors looked at the host response between 0/2/4 and 6 hr post-infection. Overall the authors want to understand how the response of DCs differ after the infection with the two strains.

The experimental design and data collection are robust and comprehensively described. The use of single-cell RNA-seq is remarkable and very well performed. This study is a first and the comparison between the two Salmonella strains brings a unique dimension to the work. The overall scope and ambition of the study is of prime interest.

This said I find the way that the authors displayed the results and subsequently the message of the publication difficult to follow. The importance of the publication is shadowed by many details and the overall quality of the paper is unfortunately decreased by figures of poor quality. Many genes names are quoted across the manuscript but how these are united in not clearly stated. Since the study is highly multidimensional (15 conditions and single-cell), and the differences across the conditions seems narrow, I understand the challenge faced by the authors.

We agree with reviewer 2 our manuscript could be better written to convey the key findings more succinctly. We have addressed this in this revised work by using Gene Ontology (GO) to identify and describe over-represented functional classes amongst differentially expressed genes. This better conveys key biological processes active in either infection and rationalizes the selection of differentially expressed genes described in the manuscript. We have also improved the quality and presentation of the data in the figures as detailed below.

At the end, how the response of DCs change between the two strains is not clear for me. How the transcriptome change between infected and bystander cells remain obscure.

Figure 3 and Figure 4 (now Figure 4 and Figure 5 in the revised manuscript) have now been improved to illustrate the main differences between the clinically invasive and non-invasive bacterial strains.

The revised Figure 4 shows that STM-D23580 bystander cells exhibit an enhanced inflammatory profile with over-expression of pro-inflammatory cytokines and transcription factors relative to STM-LT2. In contrast, the new Figure 5 focuses on the main differences between STM-D23580 and STM-LT2 infected MoDCs. In particular, MoDCs harbouring invasive *Salmonella* display differential expression of *IL10*, *MARCH1* and *CD83* to evade adaptive immune detection (Figure 6 and Figure 7 in the revised manuscript).

Considering the complex experimental design and the constraints of the journal format, we have focused the description in main text primarily on the comparison between the two *Salmonella* strains. However, we have added a new short paragraph and figure (Figure 3 in the revised manuscript) dedicated to description of the single-cell profiles observed between infected and bystander cells.

A good example that the authors could follow is Shalek et al 2014 (Nature) that has unified differential gene expression into co-regulated signatures- that could help resolving the complexity of the data and avoid list of genes. Also, instead of working with the merged data, one can consider studying the different conditions in a separated manner and only merge when needed. The authors should try to implement temporal single-cell RNA-seq analysis like monocle or diffusion maps. Also, the authors should not overstate the conclusions from the data or provide clear support for them (see below specific details).

As the reviewer accurately pointed out, our study is highly multidimensional, making it challenging to concisely illustrate the changes occurring between the different states and time points. In the revised manuscript we restrict our analysis of merged data to Figure 1. We have designed Figure 1C in the revised manuscript to illustrate an overview of the dynamics taking

place during the course of infection, highlighting the activation of cells challenged with either intracellular or extracellular stimuli in our infection model.

All subsequent figures are restricted to individual pairwise comparisons performed within individual time points. Those figures illustrate our analyses of the different conditions in a separated manner, specifically identifying differences too subtle to emerge from merged and/or pseudotemporal analyses in our data set. For instance, differences between STM-D23580 and STM-LT2 bystander cells (Figure 4 in the revised manuscript) and between STM-D23580 and STM-LT2 infected cells (Figure 5 in the revised manuscript) originate from targeted comparisons performed separately at individual time points.

1) *Figure 1 and Figure 2 poorly display the changes happening between the different conditions and should be deeply revised. In detail:*

the scheme Figure 1A indicate 2 bacteria per “infected” cells: is that true? The “Exposed” cells harbor a cell on the membrane: it is confusing because “exposed” cells should have no bacteria neither intracellular nor extracellular (as shown S-fig 2)

Figure 1A has been amended. One bacterium has been shown inside the Infected cells and the Bystander cells are not in contact with bacteria. However, extracellular *Salmonella* are shown in proximity to Bystander cells to indicate that bacteria were present in the well where these cells were cultured.

2) *Figure 1C: the heatmap in the current version cannot be read and is not informative on the changes occurring between the different states. The scale of the colour is not even indicated and the heatmap is not clustered in y axis.*

To illustrate transcriptomic changes associated with the course of infection in MoDCs, we have substituted the heat map of Figure 1C – (that previously displayed the gene expression profile of highly variable genes across all cells) – by a heat map displaying the (row-scaled) expression profile of all 342 single cells ordered in pseudotime. Pseudotime was computed using genes differentially expressed in *Salmonella*-challenged cells at 2h and 6h post-infection, to define the progress of cells through the infection model. In this way, genes progressively up or down-regulated at different rates over the course of infection partitioned into 3 co-regulated modules based on their single-cell expression profiles (Figure 1C). We also provide functional enrichment of those modules using Gene Ontology in this revised manuscript.

3) *Since the study encompass many different conditions (2 bacterial strains, 3 conditions, 4 time-points), figure 1 and figure 2 should be revised to better highlight the transcriptomics changes happening. For example, the authors could have a separate t-SNE for each bacteria strain. Figure 2A state the presence of clusters: are these clusters identified in a unsupervised manner (as written lines 128-132 and line 157-8)? The histogram fig. 2A is cryptic : how to understand such a graph? Figure 2B is extremely difficult to read. Instead, the authors should use a heatmap and this analysis seems in clear conflict with Figure 4 that address the same question (strain comparison).*

We thank the reviewer for this suggestion. We have now replaced the t-SNE plots of Figure 1B and Figure 2A-C with diffusion maps that are inherently more suitable for illustrating the progression of cells challenged with bacteria through distinct differentiation trajectories, reflecting the sequential time points of the experiment, as well as the unsupervised clusters at each time point. The heat map of Figure 2A-C in the updated manuscript displays up to 30 significant marker genes identified as DE genes ($P < 0.01$) between each unsupervised cluster and all other cells at the corresponding time point (2h, 4h and 6h p.i, respectively).

4) *The SCDE analysis comes by far too late on Figure 4 and what is exactly the difference with Figure 2B? Figure 4: what are x and y axis? The difference between the two strains is restricted to low expressed genes. Are those changes relevant statistically?*

The authors should use a heatmap so we can better look the differences between the invasive and non-invasive strain.

Figure 2 and Figure 4 (the latter being now Figure 5 in the updated manuscript) have been updated.

The SCDE package has been used for all differential expression analysis (see Method section) and it is mentioned earlier in our revised work. Moreover we have added supplementary tables documenting differential expression and gene ontology analyses performed for 1) direct comparisons between all 5 treatment groups at each time point and 2) comparisons between individual clusters and all other clusters for the identification of 'marker' genes.

Figure 5 in the revised manuscript displays all significantly differentially expressed genes between STM-LT2 infected cells and STM-D23580 infected cells. While significant GO enrichment was not found due to the limited count of differentially expressed genes, the differentially expressed gene lists were manually examined with careful review of prior literature and supporting references (e.g., the role of MARCH molecules in antigen presentation with *Salmonella* [PMID: 19666567, PMID: 27832589, PMID: 29293966, PMID: 18305173]; relevance of TFRC for *Salmonella* iron requirement [PMID: 19500110]).

5) *The authors demonstrate that DC-mediated antigen presentation to CD4+ T cells is impaired (lines 247 to 270). And consequently claim at the end that: "Our results further delineate a molecular pathway exploited by bacteria to evade immediate detection and enable dissemination in immune compromised hosts." In immune compromised hosts CD4 are dramatically reduced. So, how can such claim be made in such a strong manner?*

We have amended the main text in the following way:

"Our results further delineate a molecular pathway exploited by bacteria to evade immediate detection and enable dissemination in immune compromised hosts, in fact any negative effect on antigen presentation can further accentuate the inherent CD4+ T cell defects observed in immunocompromised individuals."

6) *Line 377-8: how can such study help designing vaccines? Is such statement supported by data?*

Our study highlights how invasive non-Typhoidal *Salmonella* STM-D23580 is capable of reducing surface expression of CD83 on MoDCs via the *IL10/MARCH1* axis, resulting in poor antigen presentation to T-cells. One could speculate that utilization of attenuated vaccines avoiding modulation of *CD83/IL10/MARCH1* gene expression in MoDCs might enhance antigen presentation. We agree the statement in the original manuscript was speculative and have removed it from the revised manuscript.

7) *Suppl Figure 11: the PCA plot does not display the hours of infection. Is it a single-time-point infection?*

Supplementary figure 11 has been moved to supplementary figure 14 in the revised manuscript. Data shown in the PCA of the new supplementary figure 14 have been sampled at 3 time points post infection (2h, 4h, 6h). The individual time points are now displayed by labels and the time progression indicated with arrows.

Minor points:

8) *In the introduction, the authors should provide the properties and some background information on STM-LT2.*

We have now stipulated the background information regarding *S. Typhimurium* LT2 in the Methods section under "bacterial strains". *S. Typhimurium* LT2 is one of the principal *Salmonella* laboratory strains used in cellular and molecular biology (McClelland et al., 2001).

9) I encourage the authors to perform an image of the infected cells using fluorescence microscopy.

We thank the reviewer for this suggestion. We have added a Supplementary figure with a fluorescence image of the infected cell cultures.

10) *Between 0 and 6hr, does the bacteria replicate in DCs? The authors should add a sentence to clarify the replication status of the bacteria.*

Under our experiment conditions, STM-LT2 and STM-D23580 demonstrate equivalent abilities to survive and to multiply within MoDCs. No significant differences were observed in the number of intracellular CFU between bacterial strains at any of the time points (Supplementary figure 4A). The percentage uptake and survival with MoDCs was also comparable for both strains with equivalent intracellular growth over time (Supplementary figure 4B, 4C). In addition, no significant differences were observed in the viability of MoDCs infected with the two bacterial strains during the course of the infection (Supplementary figure 4D).

In order to limit MOI discrepancies and to ensure consistency, all the experiments presented on this work were performed using frozen live bacterial stocks. The number of viable bacteria (CFU/ml) in each batch of stocks was carefully and frequently monitored (see Methods section).

11) *The word “exposed” should be clarified because the text hesitate between “Exposed” and “Bystander” (like in the abstract). Maybe “Bystander” is a better choice.*

We agree with the reviewer that “Bystander” is a better choice of term and it has been used to replace “Exposed” throughout the text.

12) *Page 4 line 76: is reference 22 is not related to “Salmonella growth” and should be removed from here or better explained.*

This reference was misplaced and has been removed from the revised manuscript.

13) *Suppl Figure 4: the accumulation of panels makes the figure difficult to read; one can thing maybe to have a summary figure that wraps up all the content.*

Supplementary figure 4 has been moved to supplementary figure 6. We used a heat map to represent the data in the new figure Supplementary 6 to make it easier to read.

14) *Sup Figure 4-5: what is the meaning of “feature”?*

“feature” has been converted to gene.

REVIEWERS' COMMENTS:

Reviewer #1 (Remarks to the Author):

I thank the authors for their detailed and comprehensive responses to all my comments. I have no further suggestions.

Reviewer #2 (Remarks to the Author):

In the revised manuscript, the authors have clearly improved data analysis and the overall message of the paper is much clearer. The new Figure 2A particularly attracts my attention as it demonstrates, to my knowledge, an unprecedented heterogeneity following Salmonella infection of DCs. The new Figure 3 combined with Figure 5 nicely display the transcriptomics-induced changes between LT2 and D23580.

At this stage I have only minor comments.

- Figure 2: on the heatmap: the authors could underlie the transcription factors on the heatmap with a different color
- Figure 3: make sure you have a supplementary table giving access to the gene list on the heatmap (the text does not refer to a Supplementary table)
- Typo: line 153, 160 and 176 "tables"

Response to reviewers for “Invasive *Salmonella* Exploits Divergent Immune Evasion Strategies in Infected and Bystander Dendritic Cell Subsets” Aulicino et al. 2018, NCOMMS-18-12662A.

Reviewer #1 (Remarks to the Author):

I thank the authors for their detailed and comprehensive responses to all my comments. I have no further suggestions.

Reviewer #2 (Remarks to the Author):

In the revised manuscript, the authors have clearly improved data analysis and the overall message of the paper is much clearer. The new Figure 2A particularly attracts my attention as it demonstrates, to my knowledge, an unprecedented heterogeneity following *Salmonella* infection of DCs. The new Figure 3 combined with Figure 5 nicely display the transcriptomics-induced changes between LT2 and D23580.

At this stage I have only minor comments.

- Figure 2: on the heatmap: the authors could underlie the transcription factors on the heatmap with a different color

- **Enriched GO categories and associated genes are displayed. Transcription factors are highlighted in red.**

- Figure 3: make sure you have a supplementary table giving access to the gene list on the heatmap (the text does not refer to a Supplementary table)

- **We have provided a supplementary table (supplementary table 7) with the list of genes on the heatmap of figure 3**

- Typo: line 153, 160 and 176 “tables”

- **The typos have been corrected.**